# PRUDEX-Compass: Towards Systematic Evaluation of Reinforcement Learning in Financial Markets

**Shuo Sun**  *shuo003@e.ntu.edu.sg*
*School of Computer Science and Engineering*
*Nanyang Technological University*

**Molei Qin**  *molei.qin@ntu.edu.sg*
*School of Computer Science and Engineering*
*Nanyang Technological University*

**Xinrun Wang**[*]  *xinrun.wang@ntu.edu.sg*
*School of Computer Science and Engineering*
*Nanyang Technological University*

**Bo An**[*]  *boan@ntu.edu.sg*
*School of Computer Science and Engineering*
*Nanyang Technological University*

**Reviewed on OpenReview:** *https://openreview.net/forum?id=JjbsIYOuNi*

## Abstract

The financial markets, which involve more than \$90 trillion market capitals, attract the attention of innumerable investors around the world. Recently, reinforcement learning in financial markets (FinRL) has emerged as a promising direction to train agents for making profitable investment decisions. However, the evaluation of most FinRL methods only focuses on profit-related measures and ignores many critical axes, which are far from satisfactory for financial practitioners to deploy these methods into real-world financial markets. Therefore, we introduce **PRUDEX-Compass**, which has 6 axes, i.e., Profitability, Risk-control, Universality, Diversity, rEliability, and eXplainability, with a total of 17 measures for a systematic evaluation. Specifically, i) since most existing FinRL algorithms are only designed to maximize profit with poor performance under systematic evaluation, we introduce AlphaMix+, which leverages mixture-of-experts and risk-sensitive approaches, to serve as one strong FinRL baseline that outperforms market average on all 6 axes in PRUDEX-Compass, ii) we evaluate AlphaMix+ and 7 other FinRL methods in 4 long-term real-world datasets of influential financial markets to demonstrate the usage of our PRUDEX-Compass and the superiority of AlphaMix+, iii) PRUDEX-Compass[1] together with 4 real-world datasets, standard implementation of 8 FinRL methods, a portfolio management environment and related visualization toolkits is released as public resources to facilitate the design and comparison of new FinRL methods. We hope that PRUDEX-Compass can not only shed light on future FinRL research to prevent untrustworthy results from stagnating FinRL into successful industry deployment but also provide a new challenging algorithm evaluation scenario for the reinforcement learning (RL) community.

---

[*]Corresponding Authors
[1]https://github.com/TradeMaster-NTU/PRUDEX-Compass

# 1 Introduction

Quantitative trading (QT) refers to trading strategies, which applies mathematical models to automatically identify profitable trading opportunities (Chan, 2021). With the development of the artificial intelligence, the trading volume of QT continuously increases and accounts for more than 70% and 40% trading volumes, in developed markets (e.g., US) and developing markets (e.g., China), respectively (Karpoff, 1987). How to make profitable investment decisions against the various uncertainties in quantitative trading becomes one of the main challenges for financial practitioners (An et al., 2022). Among the various machine learning methods, such as deep learning (Xu & Cohen, 2018; Sawhney et al., 2021) and boosting decision trees (Ke et al., 2017), deep reinforcement learning (DRL) is attracting increasing attention from both academia and financial industries (Sun et al., 2023) due to its stellar performance on solving complex sequential problems such as Go (Silver et al., 2017), StarCraft-II (Vinyals et al., 2019), nuclear fusion (Degrave et al., 2022) and matrix mulplication (Fawzi et al., 2022).

Deep RL has achieved significant success in various quantitative trading tasks. Specifically, FDDR (Deng et al., 2016) and iRDPG (Liu et al., 2020b) are designed to learn financial trading signals and micmic behaviors of professional traders for algorithmic trading, respectively. For portfolio management, deep RL methods are proposed to account for the impact of market risk (Wang et al., 2021b) and the commission fee (Wang et al., 2021a). A PPO-based framework (Lin & Beling, 2020) is proposed for order execution and a policy distillation mechanism is added to bridge the gap between imperfect market states and optimal execution actions (Fang et al., 2021). For market making, deep RL methods are introduced from both game-theoretic (Spooner et al., 2018) and adversarial learning (Spooner & Savani, 2020) perspectives as an adaptation of traditional mathematical models.

However, the evaluation of existing FinRL methods (Sun et al., 2022; Wang et al., 2021a; Fang et al., 2021; Liu et al., 2020a) only focuses on profit-related measures, which ignores several critical axes, such as risk-control and reliability. In addition to profitability, financial practitioners care about many other aspects of FinRL methods, i.e., how much risk I need to take for per unit of profit; how FinRL algorithms behave when the market status changes. In preliminary experiments, we find many examples that indicate the weakness of existing profit-seeking FinRL algorithms. For instance, IMIT (Ding et al., 2018) may lead to catastrophic capital loss when black swan events happen (Section 6.7 ) and SAC (Haarnoja et al., 2018) shows poor risk-control ability, which is not an ideal option for conservative traders (Section 6.3). In practice, with the existence of low signal-to-noise ratio and distribution shift in financial markets (Malkiel, 2003), FinRL methods with only great profitability performance on backtesting are very likely to overfit on historical data and become risky to be deployed in real-world scenarios (De Prado, 2018). As Robert Pardo (CEO of a hedge fund) said, he will never trade with a method that does not prove itself through a systematic evaluation (Pardo, 2011). Therefore, a benchmark for the systematic evaluation of FinRL methods is urgently needed.

In this paper, we first introduce PRUDEX-Compass, which has 6 axes with a total of 17 measures for systematic evaluation of FinRL methods. we then propose AlphaMix+, a deep RL method composed of diversified mixture-of-experts and risk-aware Bellman backup, as a strong FinRL baseline that significantly outperforms existing FinRL methods under systematic evaluation by mimicking the bottom-up hierarchical trading strategy design workflow in real-world companies (Khorana et al., 2007). In addition, we evaluate 7 widely used FinRL methods together with AlphaMix+ on 4 long-term real-world datasets spanning over 15 years on popular trading tasks to demonstrate the usage of PRUDEX-Compass and the superiority of AlphaMix+. Accompanied with an open-source library[1] of datasets, baseline implementation, RL environment and evaluation toolkits, we call for a change in how we evaluate FinRL methods to facilitate the industry deployment of FinRL methods. Moreover, PRUDEX-Compass also provides the RL community new algorithm evaluation scenarios to test the effectiveness of novel RL algorithms in the ever-changing financial markets.

# 2 PRUDEX-Compass: Systematic Evaluation of FinRL

To provide a clear exposition of FinRL evaluation, we introduce PRUDEX-Compass to provide an intuitive visual means to give financial practitioners a sense of comparability and positioning of FinRL methods. PRUDEX-Compass is composed of two central elements: i) the axis-level (inner), which specifies the different

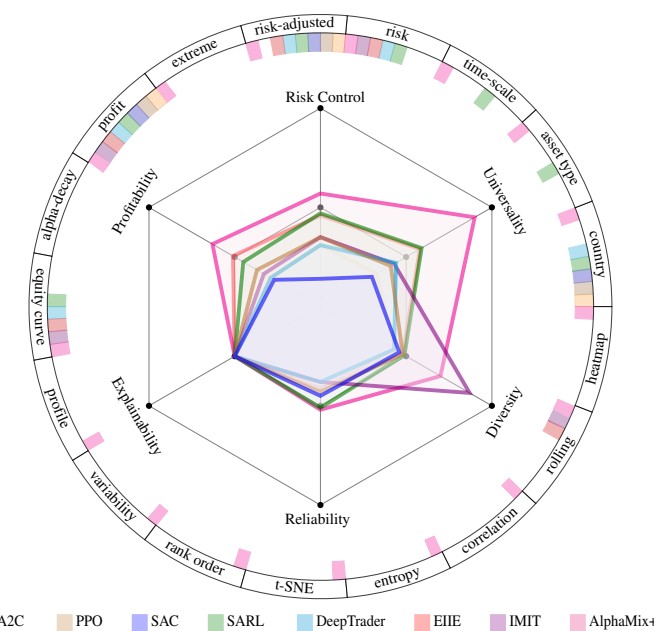

Figure 1: Illustration of our FinRL methods evaluation benchmark PRUDEX-Compass. The inner star plot provides a visual indication of the relative strength of different FinRL methods in terms of six axes. A mark on the star plot's inner circle suggests the market average[3]. The outer level of the PRUDEX-Compass specifies which particular measures have been evaluated in practice to show the status of evaluation. We fill the compass with AlphaMix+ and 7 widely-used FinRL methods to provide an intuitive example. In general, AlphaMix+ gets the best performance (largest score in 4 out of 6 axes) with a comprehensive evaluation (much more markers in the outer level).

axes considered for FinRL evaluation and ii) measure-level (outer), which specifies the measures used for benchmarking FinRL methods. Figuratively speaking, the axis-level maps out the relative strength of FinRL methods in terms of each axis, whereas the measure-level provides a compact way to visually assess which setup and evaluation measures are practically reported to point out how comprehensive the evaluation are for FinRL algorithms. To provide a practical basis, we have directly filled the exemplary compass visualization in Figure 1. The contributions of PURDEX-Compass are three-fold: i) carefully collecting 17 measures from the literature of multiple disciplines (e.g., finance, artificial intelligence, statistics and engineering) and properly categorizing them into 6 axes; ii) proposing customized version of the measures suitable for FinRL together with a set of easy-to-use visualization tools (Section 6.2–6.7); iii) introducing PRUDEX-Compass, a unified visual interface composed of two core elements to indicate the relative performance strength of FinRL methods (inner part) and their evaluation completeness (outer part). We remark that the visualization design of PRUDEX-Compass is inspired and built on top of the template code provided in CLEVA-Compass[2] (Mundt et al., 2021).

## 2.1 Axes of PRUDEX-Compass

We choose the design of the axis-level compass as a star diagram following the idea of (Wang et al., 2022; Mundt et al., 2021). Specifically, the axis-level element contains two hexagons, marks on all vertices of the two hexagons[4], and lines connecting the central point and vertices of hexagons. We add marks in the middle of central point and outer circle of the axis-level to indicate the performance of market average for convenient check. To indicate the relative strength of FinRL methods in terms of 6 critical evaluation perspectives, we first calculate the normalized score of each FinRL algorithm in terms of each axis by normalizing the numeric values of original experimental results into an integer score $t \in [0, 100]$. Inspired by the widely used performance ratio in Arcade Learning Environment (Mnih et al., 2015; Machado et al., 2018), we choose to normalize FinRL methods performance based on their relative performance to market average strategy, which is considered as the golden standard for many financial practitioners. Specifically, we assign score $t = 50$ to market average and introduce a parameter $k\%$, where $k\%$ better or worse relative to the market average is assigned 100 and 0, respectively. We further clip score values, which are larger than 100 to 100 and lower than 0 to 0, to alleviate the impact of extreme values (e.g., extremely high return rate under markets with sudden

---

[2]https://github.com/ml-research/CLEVA-Compass
[3]Market average indicates the trading strategy, which invests equal amount of money into all financial assets in the pool, to reflect the average market conditions.
[4]The edges and vertices of the inner hexagon become slightly blurry but still distinguishable after filling in the results of 8 FinRL algorithms.

increase of some Crypto assets). For the value choice of $k$, we find it is an empirically robust option to set $k = 20$ for our experiments on 8 FinRL methods across 4 financial markets. Moreover, we consult with our industry collaborators and they agree that it is reasonable to consider 20% better than the market average as a very successful trading strategy (e.g., $t = 100$). Users can directly follow our settings and compare the score (larger is better). Detailed descriptions of normalization equations are available in Appendix A.3. We then decide the position of vertices based on the score and connect them to provide a visual impression on the performance of each FinRL method. The advantages of this design are three-fold: i) it provides an ideal visual representation when comparing plot elements (Fuchs et al., 2014), ii) it allows human perceivers to quickly learn more facts by fast visual search (Elder & Zucker, 1993), and iii) the geometric region boundaries in the star plot have high priority in perception (Elder & Zucker, 1998). We introduce the 6 axis-level elements of PRUDEX-Compass as follows:

**Profitability.** Aligned with the key objective of QT, profitability focuses on the evaluation of FinRL methods' ability to gain market capital. Besides pure return, it also measures how stable (Franco & Leah, 1997) and persistent (Gârleanu & Pedersen, 2013) FinRL methods are to achieve high profit.

**Risk-Control.** Due to the well-known tradeoff between profit and risk in finance (Brink & McCarl, 1978), financial practitioners take great efforts on the assessment and control of both systematic risk and idiosyncratic risk (Goyal & Santa, 2003), which is also of vital importance in FinRL evaluation.

**Universality.** The financial market is a complex ecosystem that involves innumerable assets, countries, time-scale and trading styles. Universality tries to evaluate FinRL's ability to achieve satisfied performance (e.g., better than market average) in various quantitative trading scenarios. Designing FinRL methods with better universality (Fang et al., 2021) is in line with popular machine learning topics such as transfer learning (Pan & Yang, 2009) and meta learning (Hospedales et al., 2021).

**Diversity.** In finance, diversification refers to the process of allocating capitals in a way that reduces the exposure to any one particular asset or risk. As Markowitz (Nobel Laureate in Economics) said (Tu & Zhou, 2011), diversity is the only free lunch in investing that plays an indispensable role on enhancing profitability and risk-control. In RL community, diversity is widely used to encourage exploration (Parker et al., 2020). This axis of PRUDEX-Compass tends to address the lack of diversity evaluation of FinRL methods.

**Reliability.** RL methods have the tendency to be highly variable in performance and sensitive to many factors such as random seeds (Henderson et al., 2018) and market stationarity shift across time (Lee et al., 2010). This variability issue hinders a reliable method and can be costly or even dangerous for high-stake applications such as quantitative trading. This axis introduces techniques on RL reliability evaluation (Chan et al., 2019; Agarwal et al., 2021) with a focus on quantitative trading.

**Explainability.** Psychologically speaking, *if the users do not trust a model, they will not use it* (Ribeiro et al., 2016). Explainability generally refers to any technique that helps users or developers of models understand why models behave the way they do. In FinRL, it can come in the form that tells traders which model is effective under what market conditions or why one trading action is mistaken and how to fix it. Rigorous regulatory requirements in financial markets further enhance its importance for model debugging (Bhatt et al., 2020), monitoring (Pinto et al., 2019) and audit (Bhatt et al., 2020).

## 2.2 Measures of PRUDEX-Compass

As the inner star plot contextualizes macroscopic axes of FinRL evaluation, the outer measure-level places emphasis on detailed evaluation setup and metrics. In essence, a mark on the measure-level indicates that a method practically reports corresponding measures in its empirical investigation, where more marks indicate a more comprehensive evaluation. We list the 17 measures on the outer level of the PRUDEX-Compass in Table 1 with brief descriptions. In addition, we leave measures of FinRL explainability as future work due to the lack of FinRL algorithms with solid design of explainability. We conduct literature review on RL explainability and point their potential application in FinRL as follows. DSP (Landajuela et al., 2021) is proposed to discover symbolic policy with expert knowledge. Differentiable decision trees are incorporated into RL for better explainability (Silva et al., 2020). Another line of works tries to discover interpretable features with techniques such as self-supervised learning (Shi et al., 2020) and adversarial learning (Gupta

et al., 2020). Open-XAI (Agarwal et al., 2022) offers a comprehensive open-source framework for evaluating and benchmarking post hoc explanation methods. We plan to incorporate suitable evaluation methods, i.e., LIME (Ribeiro et al., 2016) and SHAP (Lundberg & Lee, 2017), from Open-XAI into PRUDEX-Compass with customized adoption on decision-based FinRL methods.

Table 1: Brief summary of evaluation measures in outer level of PRUDEX-Compass: Profitability, Risk-control, Universality, Diversity, rEliability, and eXplainability.

| Axes | Measures | Descriptions |
|---|---|---|
| P | Profit | A class of metrics to assess FinRL's ability to gain market capital. |
| | Alpha Decay | Loss in the investment decision making ability of FinRL methods over time due to distribution shift in financial markets (Pénasse, 2022). |
| | Equity Curve | A graphical representation of the value changes over time. |
| R | Risk | A class of metrics to assess the risk level of FinRL methods (Shiller, 1992). |
| | Risk-adjusted Profit | A class of metrics that calculate the normalized profit with regards to different kinds of risks, i.e., volatility and downside risk (Magdon & Atiya, 2004). |
| | Extreme Market | The relative performance of FinRL methods on extreme market condition during black swan events (Aven, 2013) such as war and covid-19. |
| U | Country | Financial market across both developed countries (e.g., US and Europe) and developing countries (e.g., China and India). |
| | Asset Type | Various financial asset types, i.e., stock, future, FX and Crypto |
| | Time-Scale | Both coarse-grained (e.g., day level) and fine-grained (e.g., second level) financial data to match different trading styles. |
| D | t-SNE | A statistical visualization tool to map high-dimensional time-series data points into 2-D dimension (Vander & Hinton, 2008) to assess the data-level diversity. |
| | Entropy | Entropy-based metrics from information theory (Reza, 1994) to show the diversity of FinRL methods' trading behaviors. |
| | Correlation | Metrics that account the correlation (Kirchner & Zunckel, 2011) between financial assets to assess the diversity of FinRL methods. |
| | Diversity Heatmap | A visualization tool to demonstrate the diversity of investment decisions among different financial assets with heatmap (Harris et al., 2020) |
| E | Performance Profile | A visualization of FinRL methods' empirical score distribution (Dolan & Moré, 2002), which is easy to read with qualitative comparisons. |
| | Variability | The performance standard deviation across different random seeds and hyper-parameters (Henderson et al., 2018). |
| | Rolling Window | Using rolling time window to retrain or fine-tune FinRL methods and evaluate the performance on multiple test periods (De Prado, 2018). |
| | Rank Comparison | A visualization toolkit to show the rank of FinRL methods across different metrics, which will not be dominated by extreme values (Agarwal et al., 2021). |
| X | - | We discuss current status and highlight promising further directions. |

## 3 FinRL Preliminaries and Problem Formulation

### 3.1 Portfolio Management

Portfolio management is a fundamental quantitative trading task (Sun et al., 2023), where investors hold a pool of different financial assets, i.e., stocks, bonds, as well as cash, and reallocate the proportion of capitals invested in each asset periodically to maximize future profit.

**OHLCV** refers to the raw information of bar chart acquired from the financial markets. OHLCV vector at time $t$ is denoted as $\mathbf{x}_t = (p_t^o, p_t^h, p_t^l, p_t^c, v_t)$, where $p_t^o$ is the open price, $p_t^h$ is the high price, $p_t^l$ is the low price, $p_t^c$ is the close price and $v_t$ is the volume.

**Technical Indicator** indicates high-order features derived based on a formulaic combination of the original OHLCV with financial insights. We define the vector of technical indicator at time $t$: $\mathbf{y}_t = \bigcup_k y_t^k$, where $y_t^k = f_k(\mathbf{x}_{t-h}, ..., \mathbf{x}_t, \theta^k)$, $f_k$ and $\theta^k$ are the formula function and the parameter of technical indicator $k$, respectively.

**Portfolio** is the proportion of capitals allocated to each asset that can be represented as a vector:

$$\mathbf{w_t} = [w_t^0, w_t^1, ..., w_t^M] \in R^{M+1} \quad and \quad \sum_{i=0}^M w_t^i = 1 \tag{1}$$

where $M + 1$ is the number of portfolio's constituents, including cash and $M$ financial assets. $w_t^i$ represents the ratio of the total portfolio value invested at time $t$ on asset $i$ and $w_t^0$ represents cash.

**Asset Price** refers to the vector of close price for each financial asset defined as $\mathbf{p_t} = [p_t^0, p_t^1, ..., p_t^M]$, where $p_t^i$ is the close price of asset $i$ at time $t$. Note that the price of cash $p_t^0$ is a constant.

**Portfolio Value** $v_{t+1}$ at time $t + 1$ is defined based on the asset price change and portfolio weight as:

$$v_{t+1} = v_t \sum_{i=0}^M \frac{w_t^i \ p_{t+1}^i}{p_t^i} \tag{2}$$

The objective of portfolio management is to maximize the final portfolio value given a long-term time horizon by dynamically tuning the portfolio weight at each time step. As a unified benchmark, evaluation metrics proposed in PRUDEX-Compass can be easily adopted to all quantitative trading tasks. We focus on portfolio management in this work as a demonstrative example.

## 3.2 MDP Formulation

We consider a standard RL scenario in which an agent (investor) interacts with an environment (the financial market) in discrete time. Formally, we introduce MDP, which is defined by the tuple: MDP $= (\mathcal{S}, \mathcal{A}, P, R, \gamma, H)$. Specifically, $\mathcal{S}$ is a finite set of states. $\mathcal{A}$ is a finite set of actions. $P : \mathcal{S} \times \mathcal{A} \times \mathcal{S} \longrightarrow [0, 1]$ is a state transaction function, which consists of a set of conditional transition probabilities between states. $R : \mathcal{S} \times \mathcal{A} \longrightarrow \mathcal{R}$ is the reward function, where $\mathcal{R}$ is a continuous set of possible rewards and $R$ indicates the immediate reward of taking an action in a state. $\gamma \in [0, 1)$ is the discount factor and $H$ is a time horizon indicating the length of the trading period. A (stationary) policy $\pi_\theta : \mathcal{S} \times \mathcal{A} \longrightarrow [0, 1]$, parameterized by $\theta$, assigns each state $s \in \mathcal{S}$ a distribution over actions, where $a \in \mathcal{A}$ has probability $\pi(a|s)$. A Q-value function gives expected accumulated reward when executing action $a_t$ in state $s_t$ and following policy $\pi$ in the future, which is $Q(s_t, a_t) = \mathbb{E}_{(s_{t+1}, ..., \pi)} \left[ \sum_{i=t}^T \gamma^i r(s_i, a_i) \right]$. During training, one episode corresponds to adjusting the portfolio at each time step through the whole trading periods, i.e., time scope of training set, with time horizon $H$. The objective of the agent is to learn an optimal policy: $\pi_{\theta^*} = argmax_{\pi_\theta} \mathbb{E}_{\pi_\theta} \left[ \sum_{i=t}^T \gamma^i r_{t+i} \mid s_t = s \right]$.

**State** $s_t \in \mathcal{S}$ at time $t$ involves the concatenation of technical indicator vectors of $M$ financial assets $(\mathbf{y_t^1}, \mathbf{y_t^2}, ..., \mathbf{y_t^i})$ as the external markets state, where $\mathbf{y_t^i}$ represent the technical indicator vector of asset $i$ at time $t$. In addition, a 2-dimension tuple of investors' position and remaining cash is added as internal state. **Action space** at time $t$ is an $M + 1$ dimension vector $[w_t^0, w_t^1, ..., w_t^M]$ as a portfolio $\mathbf{w}_t$ to represent the proportion of capitals invested at each asset. **Reward** $r_t$ at time $t$ is the change of portfolio value: $r_t = v_{t+1} - v_t$, where positive/negative values indicate earning/losing money, respectively.

## 3.3 Soft Actor-Critic (SAC) and Other Popular FinRL Methods

We introduce SAC (Haarnoja et al., 2018), which is the base model of many popular FinRL methods (Yuan et al., 2020). SAC is a popular off-policy actor-critic RL method based on the maximum entropy RL framework (Ziebart, 2010), which maximizes a weight objective of the reward and the policy entropy, to encourage robustness to noise and exploration. For parameter updating, SAC alternates between a soft policy evaluation and a soft policy improvement. At the soft policy evaluation step, a soft Q-function $Q_\theta(s_t, a_t)$,

which is modeled as a neural network with parameters $\theta$, is updated by minimizing the following soft Bellman residual:

$$\mathcal{L}_{critic}^{SAC}(\phi) = \mathbb{E}_{\tau_t \sim \mathcal{D}}[L_Q(\tau_t, \theta)], \tag{3}$$

$$L_Q(\tau_t, \theta) = (Q_\theta(s_t, a_t) - r_t - \gamma \bar{V}(s_{t+1}))^2, \tag{4}$$

$$where \quad \bar{V}(s_t) = \mathbb{E}_{a_t \sim \pi_\phi}[Q_{\bar{\theta}}(s_t, a_t) - \alpha \log \pi_\phi(a_t \mid s_t)]. \tag{5}$$

where $\tau_t = (s_t, a_t, r_t, s_{t+1})$ represents a transition, $\mathcal{D}$ indicates a replay buffer, $\bar{\theta}$ are the delayed parameters, and $\alpha$ is used as a temperature parameter. To conduct steps of soft policy improvement, the policy $\pi$ with its parameter $\theta$ is updated through minimizing the following objective:

$$\mathcal{L}_{actor}^{SAC}(\theta) = \mathbb{E}_{s_t \sim \mathcal{D}}[L_\pi(s_t, \phi)], \tag{6}$$

$$L_\pi(s_t, \phi) = \mathbb{E}_{a_t \sim \pi_\phi}[\alpha \log \pi_\phi(a_t \mid s_t) - Q_\theta(s_t, a_t)]. \tag{7}$$

To handle continuous action spaces, the policy is modeled as a Gaussian with mean and covariance given by neural networks. In addition to SAC, A2C (Mnih et al., 2016), a popular actor-critic RL method, shows stellar performance in algorithmic trading (Zhang et al., 2020). The simple and efficient policy gradient method PPO (Schulman et al., 2017) performs well in capturing trading opportunities for order execution (Lin & Beling, 2020). EIIE (Jiang et al., 2017) and Investor-Imitator (IMIT) (Ding et al., 2018) are two pioneering works that apply deep RL for quantitative trading. Furthermore, SARL (Ye et al., 2020) and Deeptrader (Wang et al., 2021b) are proposed with augmented market embedding to take market risk into account for portfolio management.

## 4 AlphaMix+: A Strong Baseline

Before diving into our systematic evaluation benchmark PRUDEX-Compass[5], AlphaMix+, an FinRL algorithm based on ensemble learning, is proposed to fill the gap due to the poor performance of existing FinRL methods under systematic evaluation. Considering the limitation of existing FinRL methods, the major one is that investment decisions are made by a single agent with high potential risk. The success of real-world trading firms relies on an efficient bottom-up hierarchical workflow with risk management as illustrated in Figure 2). First, multiple experts con-

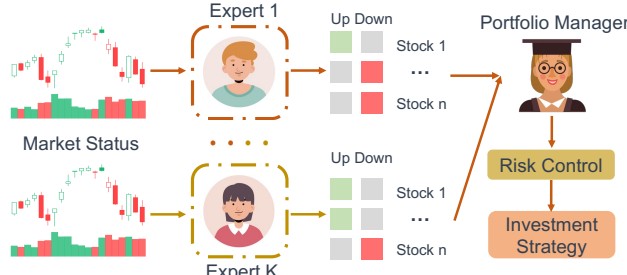

Figure 2: Workflow of real-world trading firms.

duct data analysis and build models independently based on personal trading style and risk tolerance. Later on, a senior portfolio manager summarizes their results, manage risk and makes final investment decisions.

Inspired by it, we propose AlphaMix+, a universal deep RL framework with diversified risk-aware mixture-of-experts following the idea of (Lee et al., 2021b), to mimic this efficient workflow. In principle, AlphaMix+ can be combined with most modern off-policy RL algorithms for any quantitative trading task. As SAC (Haarnoja et al., 2018) is a sample-efficiency algorithm in quantitative trading (Yuan et al., 2020), we pick it as the base model of AlphaMix+ here for exposition. An overview of AlphaMix+ is shown in Figure 3.

**Risk-aware Bellman backup.** We consider a trading firm with $N$ trading experts, i.e., $\{Q_{\theta_i}, \pi_{\phi_i}\}_{i=1}^N$, where $\theta_i$ and $\phi_i$ denote the parameters of the $i$-th experting expert's soft Q-function and policy. To apply classic Q-learning based on the Bellman backup in Eq. (3) in FinRL, one major issue is the severe negative impact of error propagation that induces the market noise to the learning trading signals (true Q-value) of

---

[5]Readers whose main interest is the evaluation benchmark may skip this Section and take AlphaMix+ as a strong FinRL baseline.

the current Q-function (Kumar et al., 2020) , which can cause unstable convergence. In order to overcome this issue, for trading expert $i$, we apply a risk-aware Bellman backup (Lee et al., 2021b) as follows:

$$\mathcal{L}_{WQ}(\tau_t, \theta_i) = w(s_{t+1}, a_{t+1})(Q_{\theta_i}(s_t, a_t) - r_t - \gamma \bar{V}(s_{t+1}))^2 \tag{8}$$

where $\tau_t = (s_t, a_t, r_t, s_{t+1})$ is a transition, $a_{t+1} \sim \pi_{\phi_i}(a \mid s_t)$, and $w(s, a)$ is a confidence weight based on the ensemble of target Q-functions:

$$w(s, a) = \sigma(-\bar{Q}_{std}(s, a) * T) + k \tag{9}$$

where $\bar{Q}_{std}(s, a)$ indicates the empirical standard deviation of all trading experts' target Q-functions $\{Q_{\bar{\theta}_i}\}_{i=1}^N$ and $T > 0$ is a temperature parameter (Hinton et al., 2015) to adapt the scale of $\bar{Q}_{std}(s, a)$. $\sigma$ is the sigmoid function and $k > 0$ is used to control the value range of confidence weight. The confidence weight is bounded in $[k, k + 0.5]$ as $\bar{Q}_{std}(s, a)$ is always a positive number. Intuitively, the objective $\mathcal{L}_{WQ}$ down weights the sample transitions with inconsistent trading suggestions from different experts (high variance across target Q-functions), resulting in a loss function for the Q-updates with better risk management.

**Diversified Experts.** We encourage the diversity between trading experts with two simple yet efficient tricks based on (Lee et al., 2021b). First, we initialize the model parameters of all trading experts with random parameter values for inducing an initial diversity in the models following (Lakshminarayanan et al., 2017; Wenzel et al., 2020). Second, we apply different samples to train each agent based on similar idea in BatchEnsemble (Wen

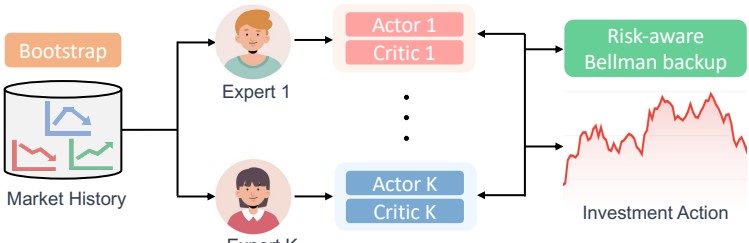

Figure 3: An illustration of AlphaMix+, a universal deep RL framework with diversified risk-aware mixture-of-experts.

et al., 2019). For each trading expert $i$ at each time step $t$, we construct the binary masks $m_{t,i}$ by sampling from the Bernoulli distribution with parameter $\beta \in (0, 1]$. Later on, we multiply the bootstrap mask to each objective function (e.g., $m_{t,i}\mathcal{L}_{\pi}(s_t, \phi_i)$ and $m_{t,i}\mathcal{L}_{WQ}(\tau_t, \theta_i)$ in Eq.(7) and Eq.(8)) while updating parameters of trading experts. This encourages each expert to think individually with diversified strategies. We find it sufficient for AlphaMix+ to have desired diversity (experiments in Section 6.6) with the two simple tricks. Other tricks such as a KL divergence (Yu et al., 2013) loss term is not further incorporated to keep simplicity. We remark that although similar diversity encouragement techniques have been shown effective in classic RL tasks (Osband et al., 2016; Lee et al., 2021b), this work is the first to explore their potential in financial markets. We conduct ablation studies on the effectiveness of each component in AlphaMix+ and parameter analysis to probe sensitivity. We put related experimental results in Appendix C.2 and C.3, respectively.

## 5 Experimental Setup

### 5.1 Datasets

We collect real-world financial datasets spanning over 15 years of US stock, China stock, Cryptocurrency (Crypto) and Foreign Exchange (FX) from Yahoo Finance and Kaggle. All raw data and related processing scripts are publicly available. We summarize statistics of the 4 datasets

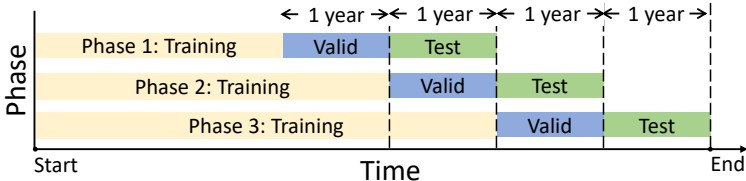

Figure 4: Train/valid/test split procedure with rolling windows.

with further elaboration in Table 2. US Stock dataset contains 10-year historical prices of 29 influential stocks with top unit price as a strong assessment of the market's overall health and tendencies. China Stock dataset contains 4-year historical prices of 47 influential stocks with top capitalization from the Shanghai exchange. Both US and China stock data is collected from Yahoo-Finance[6]. Crypto[7] dataset contains 6-year historical

---

[6]https://github.com/yahoo-finance
[7]https://www.kaggle.com/datasets/sudalairajkumar/cryptocurrencypricehistory

prices of 9 influential virtual currency with top unit price and trading volume collected. FX[8] dataset contains 20-year historical prices of 22 most popular currency with top foreign exchange reserves for US dollars. For each dataset, we filter out financial assets with missing values. For data split, we apply the similar split procedure in (Sawhney et al., 2020) with rolling window for all four datasets. As shown in Figure 4, phase 3 uses the last year for test, penultimate year for validation and the remaining of the dataset for training. For phase one and two, their validation/test sets roll back one and two years, respectively.

Table 2: Dataset statistics

| Dataset | Market | Freq | Number | Days | From | To | Source |
|---------|--------|------|--------|------|------|-----|--------|
| US Stock | US | 1d | 29 | 2517 | 12/01/03 | 21/12/31 | Yahoo |
| China Stock | China | 1h | 47 | 1036 | 16/06/01 | 20/09/01 | Yahoo |
| Crypto | - | 1d | 9 | 2014 | 16/01/01 | 21/07/06 | Kaggle |
| FX | - | 1d | 22 | 5015 | 00/01/03 | 19/12/31 | Kaggle |

## 5.2 Features

We generate 11 temporal features as shown in Table 3 to describe the stock markets following (Feng et al., 2018; Yoo et al., 2021). $z_{open}$, $z_{high}$ and $z_{low}$ represent the relative values of the open, high, low prices relative to the close price at current time step, respectively. $z_{close}$ and $z_{adj\_close}$ represent the relative values of the closing and adjusted closing prices compared with time step $t-1$. $z_{dk}$ represents a

Table 3: Features to describe the financial markets

| Features | Calculation Formula |
|----------|---------------------|
| $z_{open}, z_{high}, z_{low}$ | $z_{open} = open_t/close_t - 1$ |
| $z_{close}$ | $z_{close} = close_t/close_{t-1} - 1$ |
| $z_{d\_5}, z_{d\_10}, z_{d\_15}$ $z_{d\_20}, z_{d\_25}, z_{d\_30}$ | $z_{d\_5} = \dfrac{\sum_{i=0}^{4} close_{t-i}/5}{close_t} - 1$ |

long-term moving average of the adjusted close prices during the last $k$ time steps relative to the current close price. We apply z-score normalization on each feature.

## 5.3 Baselines

We conduct experiments with 7 representative FinRL methods including 3 classic RL methods: A2C (Mnih et al., 2016), PPO (Schulman et al., 2017), SAC (Haarnoja et al., 2018), 4 RL-based trading methods: EIIE (Jiang et al., 2017), Investor-Imitator (IMIT) (Ding et al., 2018), SARL (Ye et al., 2020) and DeepTrader (DT) (Wang et al., 2021b) and our AlphaMix+. Descriptions of baselines are as follows:

- A2C (Mnih et al., 2016) is a classic actor-critic RL algorithms that introduce an advantage function to enhance policy gradient update by reducing variance.

- PPO (Schulman et al., 2017) is a proximal policy optimization that constrain the difference between current policy and updated policy with simplified clipping term in the objective function.

- SAC (Haarnoja et al., 2018) is a widely-used off-policy actor-critic method based on the maximum entropy RL framework to encourage algorithm robustness.

- SARL (Ye et al., 2020) proposes a state-augmented RL framework, which leverages the price movement prediction as additional states, based on deterministic policy gradient (Silver et al., 2014) methods.

- DeepTrader (DT) (Wang et al., 2021b) is a policy gradient based method. To tackle the risk-return balancing issue, it simultaneously uses negative maximum drawdown and price rising rate as reward functions to balance between profit and risk with an asset scoring unit.

- EIIE (Jiang et al., 2017) is a deterministic policy gradient based RL framework, which contains: 1) an ensemble of identical independent evaluators topology; 2) a portfolio vector memory; 3) an online stochastic batch learning scheme.

---

[8]https://www.kaggle.com/datasets/brunotly/foreign-exchange-rates-per-dollar-20002019

- Investor-Imitator (IMIT) (Ding et al., 2018) imitates behaviors of different investors (e.g., oracle/collaborate/public investor) using investor-specific reward functions with a set of logic descriptors.

Non-RL methods are not included as baselines for two reasons: i) In general, RL methods outperform different types of non-RL methods (Moskowitz et al., 2012; Ke et al., 2017; Xu & Cohen, 2018) in different trading tasks. ii) PRUDEX-Compass focuses on the evaluation of RL methods in financial markets. We leave comparison with non-RL methods as future directions and discuss our plan in Section 7.

## 5.4 Training Setup

We perform all experiments on an RTX 3090 GPU with 5 fixed random seeds. We apply grid search for AlphaMix+ on Crypto and FX datasets and apply the same hyperparameters on China and US stock datasets. We try scale parameter $k$ in list $[0.3, 0.5, 0.7, 0.9]$, binomial sample parameter $\beta$ in list $[0.3, 0.4, 0.5, 0.6, 0.7]$ and temperature $T$ in list $[18, 19, 20, 21, 22]$. We explore batch size in list $[256, 512, 1024]$ and hidden size in range $[64, 128]$, We apply learning rate $7e^{-4}$ for both actor and critic. Adam is used as the optimizer. One full list of hyperparameters is available in Appendix B.1. We train AlphaMix+ for 10 epochs on all datasets. It takes about 60 minutes to train and test on China stock, US stock, FX and Crypto datasets, respectively.

For other FinRL methods, there are two conditions: i) if there are authors' official or open-source FinRL library (Liu et al., 2020a) implementations, we apply the same hyperparameters[9] for a fair comparison. This condition applies for A2C, PPO, SAC, SARL and DeepTrader. ii) if there are no publicly available implementations, we reimplement the algorithms and try our best to maintain consistency based on the original papers. This applies for EIIE and IMIT.

## 5.5 RL Environment Implementation

In this work, we apply the popular portfolio management environment (Liu et al., 2020a) implemented based on OpenAI Gym (Brockman et al., 2016), which simulates live financial markets with realistic historical market data according to the principle of time-driven simulations. During training, we feed observations of technical indicators as input of RL agents. RL agents generate a portfolio (action) and the environment returns the net value change at each time step as reward. By interacting with the environment, the trading agents will try to derive a trading strategy with high profits. In addition, the environment assumes the trading volume of agents is not very large and has little impact on the market. Then, it is reasonable to use the price fluctuation of offline historical financial data to build a model for reward calculation during online simulation. We provide a concrete example here to further clarify how FinRL environment is built. Considering a simple trading scenario with only one stock, we obtain $p_t^c$ and $p_{t+1}^c$, which denote the close price of the stock at time $t$ and $t+1$, from historical data. The action at time $t$ is to buy $k$ shares of the stock. Then, the reward $r_t$ at time $t$ is the account profit defined as $k * (p_{t+1}^c - p_t^c)$. For state, we use historical market data to calculate technical indicators in Table 3 as external state and investors' private information such as remaining cash and current position is applied as internal state. Similar procedures to build RL environment with historical market data have been applied in many FinRL work (Liu et al., 2020a; Wang et al., 2021b; Ye et al., 2020). Readers may check our open source code[1] for more details of the RL environment.

# 6 Demonstrative Usages of PRUDEX-Compass and Related Evaluation Toolkits

In this section, we conduct experiments on portfolio management with real-world datasets of 4 influential financial markets to demonstrate the usage of PRUDEX-Compass and related evaluation toolkits. In Section 6.1, we show how different investors can get a general impression on FinRL algorithms' performance with PRUDEX-Compass. Moreover, we provide example usage of other evaluation toolkits with a focus on one particular perspective including: (1) t-SNE plot to show data-level diversity, (2) PRIDE-Star to report the performance of 8 point-wise financial metrics for evaluating profitability, risk-control and diversity, (3) performance profile and rank distribution plot as unbiased and robust measures towards reliable FinRL methods, (4) portfolio diversity heatmap to evaluate the decision-level diversity, (5) extreme market scenarios

---

[9]Both authors' official or open-source FinRL library (Liu et al., 2020a) implementations are tuned in FinRL domains.

with black swan events to evaluate the risk-control and generalization ability of FinRL algorithms. In particular, investors can either use these evaluation toolkits together with PRUDEX-Compass to pursue a systematic evaluation or as an independent measure with a focus on the perspective they care about.

### 6.1 A General Impression with PRUDEX-Compass

As shown in Figure 1, we fill the PRUDEX-Compass based on the experimental results of the 8 FinRL methods. For axis-level, it directly illustrates the relative performance of each method in terms of 6 axes to provide a general impression. We normalized the score into 0 to 100 with 50 as the market average (details in Appendix A.2). For explainability, all methods are scored 50 as we leave it as future direction. AlphaMix+ performs best in all 5 remaining axes. Specifically, it outperforms other FinRL methods 53% and 43% in universality and diversity, respectively, which demonstrates the effectiveness of the weighted Bellman backup and diversified bootstrap initialization.

For measure-level, we give a mark if one measure is used in the evaluation of the FinRL methods, the goal here is to show how comprehensive the methods are evaluated. Together with all measures we proposed, AlphaMix+ clearly has a more rigorous evaluation, which makes the results more trustworthy. Arguably, PRUDEX-Compass provides a compact visualization to evaluate FinRL methods that is much better than only looking at a result table of different metrics especially when lots of FinRL methods are involved. In other words, the compass highlights the required subtleties, that may otherwise be challenging to extract from text descriptions, potentially be under-specified and prevent readers to misinterpret results based on result table display. With PRUDEX-Compass, users can flexibly pick suitable methods with regards to their personal interests. Conservative traders may prefer methods with a relative stable profit rate and low risk. Aggressive traders may pay more attention on profitability, as they are willing to take high risk to pursue extremely high profit. For international trading firms, they may have high expectation on universality and diversity.

### 6.2 Visualizing Financial Markets with t-SNE

Even though it is a wide consensus that different financial markets share different trading patterns (Campbell et al., 1998), there is a lack of visualization tool to demonstrate how different are these markets. To show data-level diversity of evaluation, we use t-SNE (Vander & Hinton, 2008) here to map all 4 datasets into a

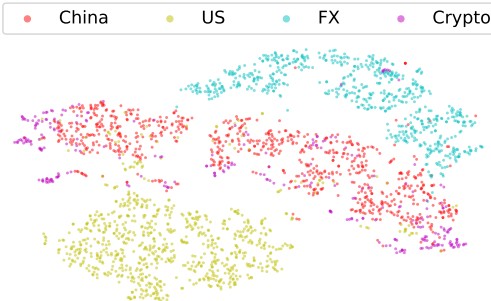

2-D dimension plot with the 11 features described in Table 3 as the input. To avoid overlapping in financial data, we pick a data point every 30 time step for each asset across 4 financial markets. Each data point corresponds to the value vector of 11 features at each time step, where necessary temporal information is maintained[10]. In Figure 5, the US stock and FX datasets lie in the lower left and upper right corner, respectively, as a whole cluster, which is consistent with their status as relative mature and stable markets (Emenyonu & Gray, 1996). For China stock and the Crypto, data points are scattered with more outliers that demonstrate their essence as emerging and violate markets (De Santis et al., 1997). The t-SNE plot is useful to provide an intuitive expression on the data-level diversity of different markets while evaluating FinRL methods.

Figure 5: t-SNE market visualization.

### 6.3 PRIDE-Star for Evaluating Profitability, Risk-Control and Diversity

As the evaluation measures for Profitability, RIsk and DivErsity (PRIDE) are point-wise metrics with real number values, we use the PRIDE-star, which is a star plot to show the relative strength of 8 metrics including 1 profit metrics: total return (TR), 2 risk metrics: volatility (Vol) (Shiller, 1992) and maximum drawdown (MDD) (Magdon & Atiya, 2004), 3 risk-adjusted profit metrics: Sharpe ratio (SR) (Sharpe, 1998), Calmar ratio (CR) and Sortino ratio (SoR), and 2 diversity metrics: entropy (ENT) (Jost, 2006) and effect number of

---

[10]It is common to incorporate temporal information into features in Fintech. For instance, $z_{d\_5}$ uses the close price at time step $t-4$ to $t$.

bets (ENB) (Kirchner & Zunckel, 2011). The mathematical definitions of these metrics are in Appendix A.1. We report the overall performance across the 4 financial markets of the 6 FinRL methods in Figure 6, where the inner circle represents market average. In general, AlphaMix+ performs best in the PRIDE-Star plot. In addition, AlphaMix+ outperforms the second best by 25% in terms of ENT that shows the effectiveness of the boostrap with random initialization component in AlphaMix+.

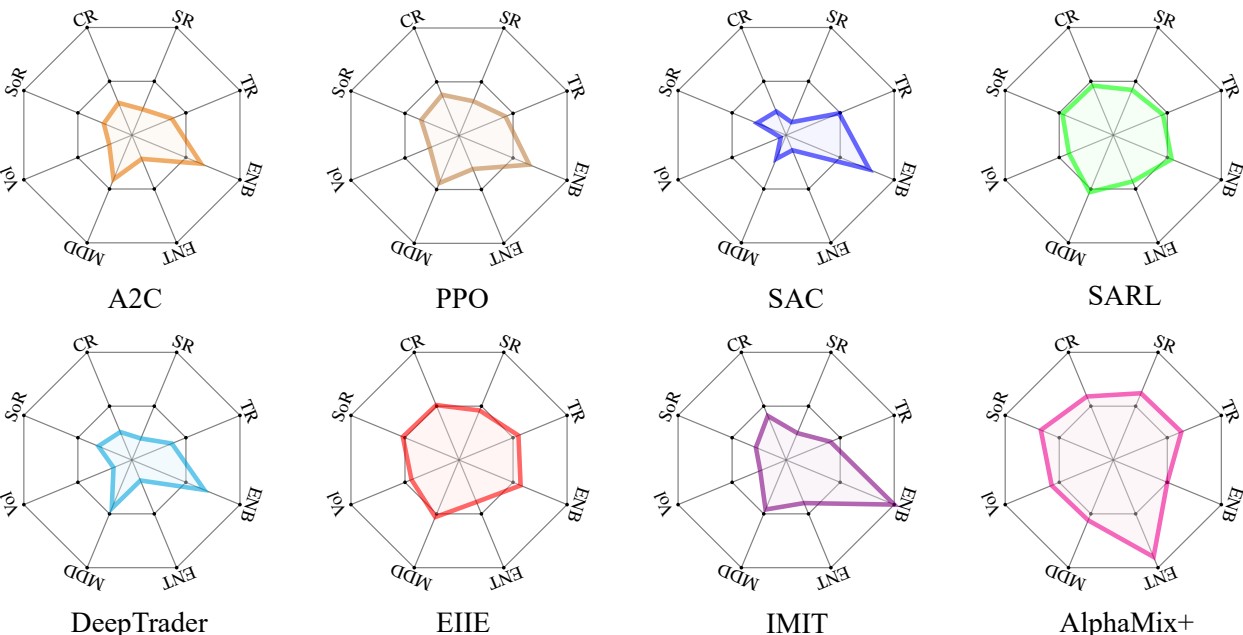

Figure 6: Overall performance across 4 financial markets on PRIDE-Star to evaluate profitability, risk-control and diversity, where AlphaMix+ achieves the best performance in 7 out of 8 metrics.

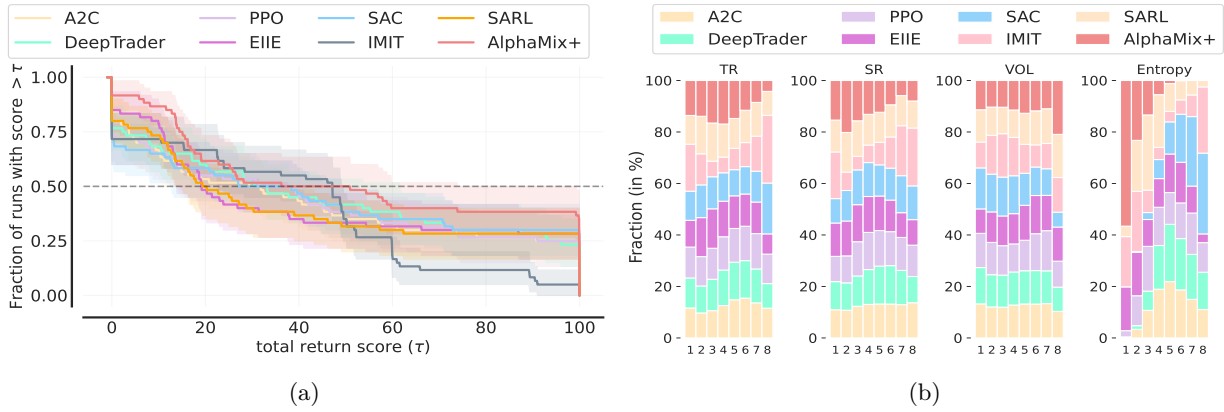

Figure 7: (a) Performance profile of total return score distributions across 4 financial markets. Shaded regions show pointwise 95% confidence bands based on percentile bootstrap with stratified sampling. (b) Rank distribution in terms of TR, SR, Vol and ENT across 4 financial markets.

## 6.4 Performance Profile: An Unbiased Approach to Report Performance

The performance profile (Agarwal et al., 2021) reports the score distribution of all runs across the 4 financial markets that are statistically unbiased and more robust to outliers compared to the widely-used mean performance. Performance profiles proposed herein visualize the empirical tail distribution function of a random score (higher curve is better), with point-wise confidence bands based on stratified bootstrap (Efron,

1979). A score distribution shows the fraction of runs above a certain normalized score that is an unbiased estimator of the underlying performance distribution. As shown in Figure 7a, AlphaMix+ is generally a robust but conservative FinRL methods that shows the least bad runs, which makes it an attractive option for conservative investors that care more about risk. However, radical investors may pick SAC as it has the largest probability of achieving score 100, which indicates a return rate higher than twice the market average.

## 6.5 Rank Distribution to Demonstrate the Rank of FinRL methods

In Figure 7b, we plot the rank distribution (Agarwal et al., 2021) of 8 FinRL methods in terms of TR, SR, VOL and Entropy across 4 financial markets with results of 5 random seeds in each market. The $i$-th column in the rank distribution plot shows the probability that a given method is assigned rank $i$ in the corresponding metrics. For x-axis, rank 1 and 8 indicate the best and worst performance.[11] For y-axis, the bar length of a given method on a given metric with rank $i$ corresponds to the % fraction of rank $i$ it achieves across the 4 financial markets, 3 test periods and 5 random seeds ($4 \times 3 \times 5 = 60$ in total), i.e., if the rank 1 column of TR is purely red, it indicates AlphaMix+ achieves the highest TR in all random seeds across 4 financial markets.

For TR and SR, AlphaMix+ slightly outperforms other methods with 27% and 35% probability to achieve top 2 performance. For Vol, SAC gets the overall best performance while AlphaMix+ goes through higher volatility. For ENT, AlphaMix+ significantly outperforms other FinRL methods with over 56% probability for rank 1, which demonstrates its ability to train mixture of diversified trading experts.

## 6.6 Visualizing Strategy Diversity with Heatmap

To demonstrate the overall investment diversity of FinRL methods, we show the average portfolio across the test period as a heatmap in Figure 8. Formally, we define the average portfolio as $\bar{w}$:

$$\bar{\mathbf{w}} = \left[ \frac{\sum_{j=t}^{t+h} w_i^0}{h+1}, \frac{\sum_{j=t}^{t+h} w_i^1}{h+1}, ..., \frac{\sum_{j=t}^{t+h} w_i^M}{h+1} \right] \tag{10}$$

where $M+1$ is the number of portfolio's constituents, including cash and $M$ financial assets. $w_t^i$ represents the ratio of the total portfolio value invested at time $t$ on asset $i$, $h$ represents the length of the evaluation period and $w_t^0$ represents cash.

For IMIT, it puts all capital in one or two assets. The portfolio of SARL and EIIE is not that diversified with near 0 weight on many assets more (red). For A2C, DT, PPO and SAC, the portfolio is closed to uniform, which is not desirable due to poor profitability. Our AlphaMix+ achieves an ideal

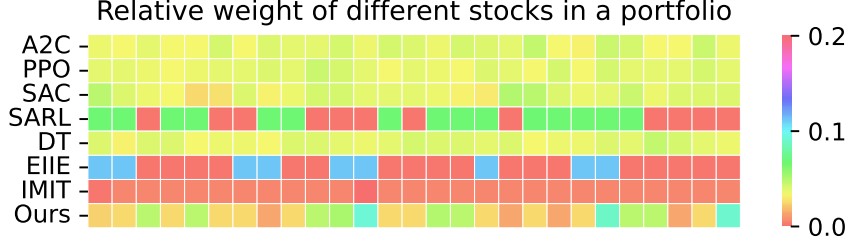

Figure 8: Heatmap of average portfolio on China stock market.

investment portfolio, which is generally diversified and allocate more weights on a few bullish stocks.

## 6.7 The Impact of Extreme Market Conditions

To further evaluate the risk-control and reliability, we pick three extreme market periods with black swan events. For China stock market, the period is from February 1st to March 31st 2021 when strict segregation policy is implemented in China to fight against the COVID-19 pandemic (Lee et al., 2021a). For US stock market, the period is from March 1st to April 30th 2020, when the global financial markets are violate due the global pandemic of COVID-19 (Mazur et al., 2021). For Crypto, the period is from April 1st to May 31st 2021 when many countries posed regulation against Crypto oligarch. To report the results of different

---

[11]For TR, SR and entropy, higher values indicate better performance. For VOL, lower values indicate better performance

metrics with different numerical value scale, we normalize them into a score $m_{score}$ as follows:

$$m_{score} = (m_{ave}/|m_{ave}|)(m_{rl}/m_{ave} - 1) * k + 1 \tag{11}$$

We define the metrics value for FinRL methods and market average as $m_{rl}$ and $m_{ave}$, respectively. $k$ is a scale parameter.

In Figure 9, we plot the bar chart of TR and SR during the period of extreme market conditions. As a conservative method, AlphaMix+'s performance is unsatisfactory in extreme market conditions, which proves the general consensus that radical methods such as DeepTrader (DT) and SARL are more suitable for extreme markets (Marimoutou et al., 2009). Analyzing the performance on extreme market conditions can shed light on the design of FinRL methods, which is in line with economists' efforts on understanding the financial markets. For instance, incorporating volatility-aware auxiliary task (Sun et al., 2022) and multi-objective RL (Hayes et al., 2022) in AlphaMix+ may further make it be aware of extreme market conditions in advance and behave as a profit-seeking agents to achieve better performance during extreme market conditions.

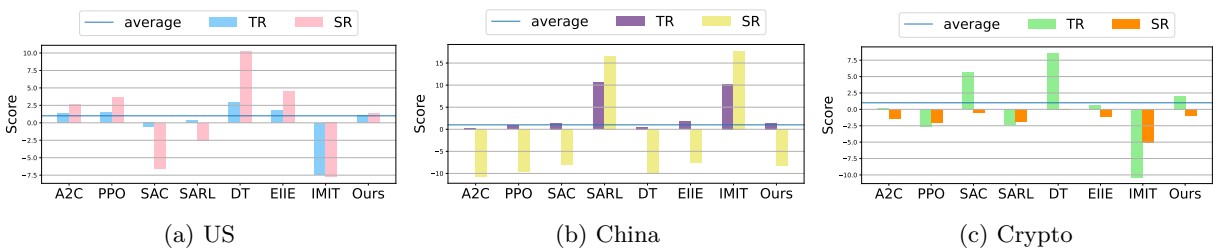

Figure 9: Performance of FinRL methods during extreme market conditions.

## 7 Discussion

**Complementary Related Efforts.** Apart from the above aspects described in PRUDEX-Compass, there also exist several orthogonal perspectives for FinRL evaluation, which encompass a *check-list* (Pineau et al., 2021) for quantitative experiments, the construction of elaborate *dataset sheets* (Gebru et al., 2021), and the creation of *model cards* (Mitchell et al., 2019). We stress that these perspectives remain indispensable, as novel datasets and their variants are regularly suggested in FinRL and the PRUDEX-Compass does not disclose intended use with respect to human-centered application domains, ethical considerations, or their caveats. We believe that it is best to report both the prior works and PRUDEX-Compass together to further improve the evaluation quality.

**Potential Impact.** For FinRL community, PRUDEX-Compass can serve as i) a systematic evaluation toolkit; ii) a benchmark for comprehensive comparison of FinRL methods and iii) a standard code base for future design and development of novel FinRL algorithms. We hope that PRUDEX-Compass could encourage both researchers and financial practitioners to avoid fooling themselves by evaluating FinRL methods in a systematic way and facilitate the design of stronger FinRL methods. While accounting for all elements in PRUDEX-Compass is not a panacea, we believe it is a good start with more trustworthy results for the community and further increase the confidence for real-world industry deployment. For RL community, PRUDEX-Compass introduces a new challenging scenario with different evaluation axes for the test of novel RL algorithms in financial market. For FinTech community, this work enables industry practitioners with limited RL background to easily explore the potential of RL in different FinTech scenarios. In addition, the usage of PRUDEX-Compass is not limited in RL settings, most elements of PRUDEX-Compass can be easily generalized to supervised learning settings with broader impact.

**Future Plans.** We plan to improve PRUDEX-Compass from the following perspectives: i) For axis-level, we plan to explore the evaluation of FinRL explainability with measures and plots; ii) For measure-level, we plan to include metrics to evaluate alpha decay and more metrics, e.g., optimality gaps, for profits and risks; iii) To further improve the accuracy and reliability of PRUDEX-Compass, we plan to customize existing scientific and non-exploitative RL evaluation procedures (Jordan et al., 2020) into FinRL domains

for the normalization and summarization of experimental results. iv) For a comprehensive evaluation under different market stationarity assumptions, we plan to add one toolkit to automatically categorize market into different styles (e.g., bull/bear) and evaluate the performance of FinRL methods under different styles. Furthermore, there can be a data-driven simulator to generate unseen stylized data for further evaluation; v) For visualization, we plan to further develop a GUI software version accompanied with a website to further lower the barrier for dissemination and use; vi) As most axes and measures in PRUDEX-Compass are also key points for non-RL trading scenarios, we plan to bring PRUDEX-Compass into more general machine learning settings with implementations and results of non-RL methods for broader impact.

**Auxiliary Experiments.** Due to space limitations, we have included some auxiliary yet important experiments in Appendix C. Specifically, the result tables with mean and standard deviation of the 8 metrics in PRIDE-Star are reported in Appendix C.1. We plot the PRIDE-Star, performance profile and rank comparison of each financial market individually in Appendix C.4, C.5, C.6, respectively.Furthermore, we include ablation studies on the effectiveness of each component in AlphaMix+ in Appendix C.2 and hyperparameter sensitivity experiments in Appendix C.3.

**Hosting, Maintenance, Licensing.** The PRUDEX-Compass datasets are hosted on Google Drive. The source code are publicly available at https://github.com/TradeMaster-NTU/PRUDEX-Compass. The authors will provide important bug fixes to the community as commits to the GitHub repository. There will be summary of changes to the code and the datasets in the README web page of the GitHub repository. In the unlikely case that the Google Drive link stops operating, we will migrate the dataset to another hosting and announce the new links in the GitHub repository. The provided source code and dataset are copyrighted by us and under the MIT license[12]. Users have the permission to reuse the codes for any purpose.

# 8 Acknowledgments

This project is supported by the National Research Foundation, Singapore under its Industry Alignment Fund – Pre-positioning (IAF-PP) Funding Initiative. Any opinions, findings and conclusions or recommendations expressed in this material are those of the author(s) and do not reflect the views of National Research Foundation, Singapore.

---

[12]https://opensource.org/licenses/MIT

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

## Appendix

## A    PRUDEX-Compass

### A.1    Definition of Financial Metrics

**Profit** measure contains metrics to evaluate FinRL methods' ability to gain market capital. Total return (TR) is the percent change of net value over time horizon $h$. The formal definition is $TR = (n_{t+h} - n_t)/n_t$, where $n_t$ is the corresponding value at time $t$.

**Risk** includes a class of metrics to assess the risk level of FinRL methods.

- **Volatility (Vol)** is the variance of the return vector $\mathbf{r}$. It is widely used to measure the uncertainty of return rate and reflects the risk level of strategies. The definition is $Vol = \sigma[\mathbf{r}]$.

- **Maximum drawdown (MDD)** measures the largest decline from the peak in the whole trading period to show the worst case. The formal definition is $MDD = \max_{\tau \in (0,t)}[\max_{t \in (0,\tau)} \frac{n_t - n_\tau}{n_t}]$.

- **Downside deviation (DD)** refers to the standard deviation of trade returns that are negative.

**Risk-adjusted Profit** calculates the potential normalized profit by taking one share of the risk. We define three metrics with different types of risk:

- **Sharpe ratio (SR)** refers to the return per unit of deviation: $SR = \frac{\mathbb{E}[\mathbf{r}]}{\sigma[\mathbf{r}]}$.

- **Sortino ratio (SoR)** is another risk-adjusted profit, which applies DD as risk measure: $SoR = \frac{\mathbb{E}[\mathbf{r}]}{DD}$.

- **Calmar ratio (CR)** is another risk-adjusted profit, which applies MDD as risk measure: $CR = \frac{\mathbb{E}[\mathbf{r}]}{MDD}$.

### A.2    Creating a PRUDEX-Compass

To make the PRUDEX-Compass as accessible as possible and disseminate in a convenient way, we provide two options for practical use based on the template provided in CLEVA-Compass (Mundt et al., 2021).

- We provide a **LaTeX template** for PRUDEX-Compass, making use of the TikZ library to draw the compass with. We envision that such a template makes it easy for readers to include a compass into their future research, where they can adapt the naming and values of the entries respectively.

- We further provide a **Python script** to generate the PRUDEX-Compass. In fact, because the use of drawing in LaTeX with TikZ may be unintuitive for some, we have written a Python script that automatically fills above LaTeX template, so that it can later simply be included into a LaTeX document. The Python script takes a path to a JSON file that needs to be filled by the user. There is a default JSON file that is easy to adopt.

### A.3    Computing Scores for PRUDEX-Compass Axes

We introduce how the value of each axe for each FinRL method is computed in this subsection. The basic principle is to propose a distinguishable and robust way to show the performance difference of FinRL methods across multiple evaluation measures in terms of each axe. Generally speaking, we normalize the performance of different measures on the 6 axes to a score from 0 to 100. We mark market average strategy (evenly invest on all assets) as 50. All scores are calculated with the average of 4 financial markets. We define the metrics value for FinRL methods and market average as $m_{rl}$ and $m_{ave}$, respectively. For the 4 profit-related metrics (TR, SR, CR, SoR), we normalized them into a score $S_{pro}$ with range $[0, 100]$: $S_{pro} = (m_{rl}/m_{ave} - 0.8) * 250$, where 20% higher profit than market average is scored 100. We clip values lower than 0 and higher than 100 as 0 and 100, respectively. For the 2 risk metrics (Vol, MDD), we normalized them into a score $S_{risk}$ with

range $[0, 100]$: $S_{risk} = (1.2 - m_{rl}/m_{ave}) * 250$, where 20% lower risk than market average is scored 100. We clip values lower than 0 and higher than 100 as 0 and 100, respectively.

For universality, we directly use the raw return rate and calcualte the 4 indicators of profitability, we then plot a rank graph and for each measures of profitability, we multiply the probability obtained from the rank matrix and the rankscore(if it's 1st, the score is 100 and if 6th, the score is 0) to get a score for that measure. Then we average the 4 measures together to get the university score for that algorithm. For the 2 diversity metrics (ENT, ENB), we normalized them into a score $S_{div}$:

$$S_{div} = \begin{cases} m_{rl}/m_{ave} \times 100, & \text{for ENT} \\ m_{rl}/m_{ave} \times 50, & \text{for ENB} \end{cases}$$

where the diversity of uniform policy is scored 100 for ENT and 50 for ENB. For explainability, we set the explainability score 50 for all FinRL methods. For reliability, we use the total return rate score we just normalized using average policy as a indicator and draw a performance profile graph, then we use the area under the curve for each algorithm to calculate the reliability. The constants in the equations (e.g., 20%) are applied to make the plot distinguishable and easy to follow, which does not influence the robustness of axes in PRUDEX-Compass.

## B   Experiment Setup

### B.1   Training Setup

Table 4: Hyperparameters of AlphaMix+

| Hyperparameter | Value | Hyperparameter | Value |
|---|---|---|---|
| Replay buffer size | 10000 | Initial step | 10000 |
| Layer(MLP) | (128,128) | Stacked frame | 3 |
| Evaluation episodes | 10 | Optimizer | Adam |
| Temperature | 20 | Uncertainty | 0.5 |
| Actor learning rate | 0.0007 | Critic learning rate | 0.0007 |
| Batch size | 256 | Action numbers | 29(US) 47(China) 22(FX) 9(Crypto) |
| Discount $\gamma$ | 0.99 | Ber_mean | 0.5 |
| Non-linear | Sigmoid | Observation | Number of assets $\times$ 11 |

## C   Experimental Results

### C.1   Result Table

In this subsection, we report detailed results of 8 metrics in the four financial markets. Since we apply 1 year rolling window during training, each financial market has 3 tables for 3 consecutive years.

Table 5: US Stock 2021

| Metrics | A2C | PPO | SAC | SARL | DeepTrader | EIIE | IMIT | AlphaMix+ |
|---|---|---|---|---|---|---|---|---|
| TR(%) | 12.4±6.3 | 13.1±4.4 | 20.0±10.3 | 17.3±2.0 | 17.2±7.4 | 16.9±1.2 | 4.0±10.1 | 20.7±1.8 |
| SR | 1.03±0.50 | 0.94±0.26 | 1.30±0.57 | 1.37±0.12 | 1.22±0.50 | 1.39±0.08 | 0.35±0.64 | 1.64±0.11 |
| CR | 0.81±0.34 | 0.83±0.13 | 0.91±0.19 | 1.06±0.03 | 0.84±0.23 | 1.06±0.02 | 0.47±0.61 | 1.09±0.02 |
| SoR | 1.52±0.70 | 1.38±0.44 | 1.89±0.87 | 2.02±0.21 | 1.81±0.75 | 2.03±0.12 | 0.52±0.91 | 2.36±0.17 |
| MDD(%) | 15.0±1.9 | 15.3±2.5 | 19.7±4.6 | 15.6±1.2 | 19.1±3.9 | 15.2±0.6 | 14.6±3.7 | 17.9±1.4 |
| VOL(%) | 0.78±0.06 | 0.87±0.03 | 0.91±0.02 | 0.76±0.02 | 0.86±0.03 | 0.73±0.03 | 1.0±0.12 | 0.74±0.02 |
| ENT | 2.26±0.26 | 1.82±0.02 | 1.67±0.02 | 2.79±0.11 | 1.82±0.01 | 2.90±0.26 | 1.89±0.1 | 3.25±0.12 |
| ENB | 1.34±0.10 | 1.49±0.01 | 1.61±0.02 | 1.14±0.06 | 1.51±0.02 | 1.19±0.10 | 1.73±0.10 | 1.11±0.03 |

Table 6: US Stock 2020

| Metrics | A2C | PPO | SAC | SARL | DeepTrader | EIIE | IMIT | AlphaMix+ |
|---------|-----|-----|-----|------|-----------|------|------|-----------|
| TR(%) | 7.46±4.86 | 18.3±8.4 | 7.77±21.1 | 9.23±8.79 | 10.2±10.5 | 15.8±16.1 | -20.6±9.5 | 12.7±1.7 |
| SR | 0.36±0.12 | 0.62±0.19 | 0.37±0.52 | 0.41±0.21 | 0.42±0.25 | 0.55±0.30 | -0.28±0.21 | 0.52±0.04 |
| CR | 0.36±0.12 | 0.54±0.15 | 0.30±0.48 | 0.38±0.19 | 0.42±0.22 | 0.49±0.22 | -0.28±0.22 | 0.47±0.03 |
| SoR | 0.44±0.14 | 0.76±0.23 | 0.50±0.63 | 0.50±0.26 | 0.56±0.33 | 0.68±0.38 | -0.36±0.27 | 0.62±0.05 |
| MDD(%) | 36.6±2.4 | 42.2±2.4 | 42.8±5.3 | 37.8±3.19 | 34.9±3.91 | 38.9±6.69 | 43.5±6.42 | 38.2±0.96 |
| VOL(%) | 2.25±0.09 | 2.32±0.03 | 2.45±0.16 | 2.26±0.17 | 2.31±0.09 | 2.21±0.23 | 2.9± 0.39 | 2.16±0.04 |
| ENT | 1.96±0.20 | 1.82±0.02 | 1.61±0.03 | 2.07±0.67 | 1.82±0.02 | 2.41±0.67 | 1.85 ±0.21 | 3.22±0.03 |
| ENB | 1.05±0.01 | 1.05±0.004 | 1.1±0.003 | 1.05±0.03 | 1.05±0.002 | 1.05±0.05 | 1.13± 0.04 | 1.0± 0.006 |

Table 7: US Stock 2019

| Metrics | A2C | PPO | SAC | SARL | DeepTrader | EIIE | IMIT | AlphaMix+ |
|---------|-----|-----|-----|------|-----------|------|------|-----------|
| TR(%) | 20.5±9.27 | 21.5±10.1 | 27.3±9.79 | 27.7±1.63 | 22.9±3.51 | 30.2±8.42 | 20.6±9.52 | 25.1±1.42 |
| SR | 1.47±0.58 | 1.53±0.62 | 1.72±0.54 | 2.00±0.15 | 1.60±0.24 | 2.12±0.27 | 1.28±0.21 | 1.98±0.06 |
| CR | 1.00±0.06 | 1.01±0.06 | 1.05±0.09 | 1.02±0.05 | 1.01±0.05 | 1.06 ±0.05 | 0.28±0.22 | 1.03±0.007 |
| SoR | 1.96±0.78 | 2.04±0.83 | 2.17±0.78 | 2.52±0.24 | 2.16±0.43 | 2.82±0.42 | 0.36±0.27 | 2.47±0.09 |
| MDD(%) | 18.8±5.95 | 19.4±6.26 | 23.4±4.70 | 24.7±2.63 | 21.2±2.41 | 25.2±4.83 | 43.6±6.42 | 22.3±0.99 |
| VOL(%) | 0.82±0.01 | 0.83±0.02 | 0.91±0.05 | 0.79±0.10 | 0.84±0.03 | 0.79±0.11 | 1.9±0.39 | 0.73±0.02 |
| ENT | 1.83±0.01 | 1.82±0.01 | 1.63±0.02 | 2.08±0.67 | 1.81±0.01 | 2.32±0.48 | 1.89±0.16 | 3.27±0.03 |
| ENB | 1.28±0.01 | 1.27±0.01 | 1.38±0.01 | 1.29±0.21 | 1.26±0.005 | 1.15±0.06 | 1.73±0.10 | 1.05± 0.01 |

Table 8: China Stock 2020

| Metrics | A2C | PPO | SAC | SARL | DeepTrader | EIIE | IMIT | AlphaMix+ |
|---------|-----|-----|-----|------|-----------|------|------|-----------|
| TR(%) | 5.52±4.82 | 6.20±7.57 | 13.4±20.1 | 11.4±8.84 | 12.3±14.2 | 13.0±3.8 | 52.6±20.8 | 14.8±3.30 |
| SR | 0.35±0.21 | 0.40±0.37 | 0.59±0.68 | 0.63±0.41 | 0.62±0.60 | 0.73± 0.17 | 2.68±0.83 | 0.79±0.12 |
| CR | 0.25±0.15 | 0.29±0.27 | 0.41±0.48 | 0.42±0.27 | 0.36±0.39 | 0.51±0.08 | 1.18±0.24 | 0.53±0.06 |
| SoR | 0.40±0.24 | 0.45±0.42 | 0.71±0.81 | 0.71±0.46 | 0.71±0.69 | 0.80±0.18 | 4.04±1.2 | 0.89±0.15 |
| MDD(%) | 28.1±3.68 | 25.3±1.59 | 29.9±7.38 | 26.5±5.09 | 31.5±6.02 | 27.0±2.41 | 34.2±9.16 | 29.1±1.85 |
| VOL(%) | 0.74±0.04 | 0.68±0.004 | 0.81±0.06 | 0.68±0.02 | 0.76±0.01 | 0.67±0.02 | 0.66±0.09 | 0.69±0.01 |
| ENT | 1.85±0.42 | 2.82±0.01 | 1.10±0.02 | 2.60±0.17 | 1.53±0.001 | 2.39±0.10 | 1.30 ±0.85 | 3.12±0.02 |
| ENB | 1.12±0.05 | 1.04±0.01 | 1.27±0.01 | 1.04±0.01 | 1.16±0.004 | 1.06± 0.01 | 2.82± 0.85 | 1.02±0.01 |

Table 9: China Stock 2019

| Metrics | A2C | PPO | SAC | SARL | DeepTrader | EIIE | IMIT | AlphaMix+ |
|---------|-----|-----|-----|------|-----------|------|------|-----------|
| TR(%) | 31.9±10.6 | 29.7±8.42 | 25.4±10.4 | 32.6±5.78 | 22.1±13.6 | 36.2± 7.09 | -7.14±0.82 | 32.2±2.20 |
| SR | 1.59±0.46 | 1.65±0.39 | 1.13±0.38 | 1.80±0.24 | 1.19±0.59 | 1.77± 0.05 | -0.34±0.1 | 1.79±0.10 |
| CR | 0.90±0.16 | 0.94±0.13 | 0.79±0.23 | 1.01±0.07 | 0.74±0.21 | 1.04± 0.05 | -0.29±0.07 | 1.01±0.02 |
| SoR | 2.32±0.75 | 2.41±0.63 | 1.70±0.64 | 2.63±0.34 | 1.72±0.91 | 2.57± 0.15 | -0.37± 0.08 | 2.62±0.14 |
| MDD(%) | 31.7±2.85 | 28.6±2.82 | 29.8±2.38 | 28.9±2.38 | 27.5±4.92 | 30.5±3.74 | 20.4± 0.63 | 28.9±0.90 |
| VOL(%) | 0.65±0.02 | 0.58±0.004 | 0.74±0.02 | 0.57±0.01 | 0.63±0.02 | 0.64±0.11 | 0.77± 0.06 | 0.57±0.01 |
| ENT | 1.54±0.01 | 2.85±0.005 | 1.02±0.02 | 2.47±0.17 | 1.53±0.009 | 1.97±0.90 | 1.30±0.85 | 3.15±0.07 |
| ENB | 1.18±0.009 | 1.05±0.002 | 1.32±0.009 | 1.05±0.01 | 1.18±0.004 | 1.21 ±0.18 | 1.19±0.85 | 1.04±0.01 |

Table 10: China Stock 2018

| Metrics | A2C | PPO | SAC | SARL | DeepTrader | EIIE | IMIT | AlphaMix+ |
|---------|-----|-----|-----|------|-----------|------|------|-----------|
| TR(%) | -6.86±11.30 | -6.56±3.66 | -4.05±10.77 | -5.77±4.04 | 5.67±5.95 | -2.27±1.99 | -7.14± 0.82 | -0.70±1.38 |
| SR | -0.28±0.61 | -0.31±0.22 | -0.12±0.50 | -0.25±0.21 | 0.37±0.30 | -0.05± 0.12 | -0.34±0.1 | 0.03±0.08 |
| CR | -0.14±0.50 | -0.25±0.16 | -0.06±0.42 | -0.20±0.15 | 0.40±0.32 | -0.04±0.11 | -0.29±0.07 | 0.04±0.10 |
| SoR | -0.33±0.78 | -0.41±0.26 | -0.15±0.75 | -0.35±0.29 | 0.54±0.45 | -0.07± 0.17 | -0.37±0.08 | 0.04±0.11 |
| MDD(%) | 22.3±5.52 | 20.5±1.55 | 25.9±4.95 | 20.2±3.49 | 19.1±3.39 | 17.5±1.52 | 20.4±0.63 | 16.1±1.45 |
| VOL(%) | 0.67±0.03 | 0.59±0.01 | 0.75±0.03 | 0.61±0.03 | 0.66±0.02 | 0.59± 0.03 | 0.77±0.06 | 0.57±0.02 |
| ENT | 1.54±0.01 | 2.85±0.007 | 1.01±0.03 | 2.41±0.40 | 1.53±0.007 | 2.55±0.15 | 1.30± 0.85 | 3.12±0.02 |
| ENB | 1.31±0.01 | 1.08±0.005 | 1.57±0.009 | 1.11±0.10 | 1.30±0.009 | 1.05± 0.01 | 1.19 ±0.08 | 1.03±0.003 |

Table 11: Crypto 2021

| Metrics | A2C | PPO | SAC | SARL | DeepTrader | EIIE | IMIT | AlphaMix+ |
|---|---|---|---|---|---|---|---|---|
| TR(%) | 223±346 | 146±129 | 377±982 | 199±195 | 398±344 | 146±137 | 140± 155 | 291±71 |
| SR | 1.38±0.74 | 1.22±0.56 | 1.13±0.67 | 1.41±0.45 | 1.71±0.45 | 1.46± 0.42 | 1.24±0.34 | 1.87±0.09 |
| CR | 1.77±1.02 | 1.48±0.53 | 2.18±2.05 | 1.75±0.74 | 2.18±0.84 | 1.44±0.49 | 1.39±0.58 | 2.02±0.26 |
| SoR | 2.15±1.23 | 2.13±1.11 | 3.16±3.31 | 2.42±1.24 | 2.97±1.35 | 2.42± 1.22 | 2.03± 0.80 | 3.14±0.56 |
| MDD(%) | 77.1±15.7 | 74.9±10.4 | 80.1±15.6 | 79.4±10.0 | 85.1±8.42 | 72.5±10.2 | 71.0± 11.2 | 85.7±4.14 |
| VOL(%) | 7.30±1.91 | 6.97±0.94 | 10.94±8.13 | 7.26±2.31 | 7.85±2.05 | 5.22±1.06 | 5.56± 1.61 | 6.80±0.84 |
| ENT | 1.02±0.31 | 0.72±0.05 | 0.47±0.07 | 1.42±0.25 | 0.56±0.04 | 1.53±0.23 | 0.58±0.23 | 2.09±0.02 |
| ENB | 1.79±0.15 | 1.94±0.04 | 1.99±0.05 | 1.67±0.40 | 1.97±0.03 | 1.12±0.05 | 1.66±0.38 | 1.47±0.15 |

Table 12: Crypto 2020

| Metrics | A2C | PPO | SAC | SARL | DeepTrader | EIIE | IMIT | AlphaMix+ |
|---|---|---|---|---|---|---|---|---|
| TR(%) | 61.4±58.1 | 59.1±58.4 | 16.0±38.4 | 31.4±12.8 | 37.9±27.1 | 30.7± 11.7 | 55.3± 17.6 | 33.8±6.16 |
| SR | 1.12±0.66 | 1.12±0.67 | 0.51±0.50 | 0.81±0.24 | 0.85±0.38 | 0.79±0.17 | 1.03± 0.08 | 0.86±0.08 |
| CR | 1.06±0.47 | 1.02±0.45 | 0.59±0.58 | 0.86±0.23 | 0.97±0.42 | 0.82±0.19 | 0.73± 0.07 | 0.87±0.09 |
| SoR | 1.58±1.20 | 1.62±1.22 | 0.59±0.63 | 0.89±0.29 | 1.04±0.54 | 0.92±0.18 | 1.49± 0.32 | 0.94±0.10 |
| MDD(%) | 52.9±7.58 | 52.2±8.03 | 55.7±2.12 | 46.6±8.08 | 50.3±8.04 | 44.6±3.22 | 56.6± 4.43 | 46.5±2.43 |
| VOL(%) | 3.86±0.22 | 3.75±0.17 | 4.46±0.35 | 3.66±0.76 | 4.05±0.32 | 3.37± 0.36 | 2.86± 0.53 | 3.46±0.22 |
| ENT | 0.59±0.04 | 0.59±0.04 | 0.45±0.02 | 1.32±0.45 | 0.60±0.01 | 1.24±0.52 | 0.58±0.52 | 2.17±0.07 |
| ENB | 1.05±0.01 | 1.06±0.03 | 1.07±0.03 | 1.01±0.005 | 1.05±0.01 | 1.01±0.01 | 1.01±0.01 | 1.00±0.001 |

Table 13: Crypto 2019

| Metrics | A2C | PPO | SAC | SARL | DeepTrader | EIIE | IMIT | AlphaMix+ |
|---|---|---|---|---|---|---|---|---|
| TR(%) | 82.8±51.5 | 85.8±57.1 | 73.9±33.9 | 61.0±19.3 | 39.1±54.5 | 50.8±13.2 | 110.6± 17.6 | 68.3±17.8 |
| SR | 1.37±0.50 | 1.39±0.55 | 1.27±0.35 | 1.26±0.26 | 0.83±0.59 | 1.15±0.20 | 2.06± 0.08 | 1.43±0.22 |
| CR | 1.14±0.30 | 1.14±0.33 | 1.06±0.24 | 1.06±0.11 | 0.73±0.39 | 0.98±0.10 | 1.45± 0.07 | 1.08±0.11 |
| SoR | 2.19±0.98 | 2.22±1.02 | 2.10±0.65 | 1.89±0.39 | 1.30±1.12 | 1.77±0.28 | 2.99± 0.32 | 2.16±0.35 |
| MDD(%) | 59.5±9.64 | 59.9±10.5 | 59.1±7.74 | 52.4±6.49 | 51.3±11.3 | 49.9±5.80 | 56.6± 4.43 | 55.5±3.58 |
| VOL(%) | 3.69±0.36 | 3.69±0.35 | 3.63±0.27 | 3.29±0.35 | 3.46±0.24 | 3.15±0.30 | 2.86± 0.53 | 3.09±0.18 |
| ENT | 0.60±0.02 | 0.62±0.03 | 0.43±0.03 | 1.21±0.40 | 0.59±0.02 | 1.24±0.52 | 0.58±0.52 | 2.15±0.10 |
| ENB | 1.15±0.006 | 1.14±0.01 | 1.15±0.008 | 1.06±0.02 | 1.14±0.01 | 1.01±0.01 | 1.01±0.01 | 1.01±0.01 |

Table 14: Foreign Exchange 2019

| Metrics | A2C | PPO | SAC | SARL | DeepTrader | EIIE | IMIT | AlphaMix+ |
|---|---|---|---|---|---|---|---|---|
| TR(%) | -0.94±3.14 | -1.49±2.68 | -1.02±3.11 | 0.96±0.36 | -0.19±2.88 | 1.42±0.24 | -5.53± 0.05 | 1.22±0.46 |
| SR | -0.22±0.73 | -0.34±0.64 | -0.21±0.64 | 0.29±0.09 | -0.03±0.66 | 0.39±0.05 | -0.97± 0.01 | 0.41±0.16 |
| CR | -0.07±0.42 | -0.16±0.35 | -0.02±0.43 | 0.22±0.08 | 0.04±0.46 | 0.32±0.07 | -0.49± 0.01 | 0.36±0.17 |
| SoR | -0.32±1.05 | -0.51±1.00 | -0.26±0.86 | 0.50±0.17 | 0.06±1.11 | 0.65±0.07 | -1.5± 0.02 | 0.74±0.31 |
| MDD(%) | 7.90±1.85 | 6.28±1.37 | 6.94±2.07 | 4.60±0.47 | 6.84±1.38 | 4.65± 0.43 | 11.17± 0.05 | 3.77±0.56 |
| VOL(%) | 0.27±0.01 | 0.28±0.01 | 0.31±0.01 | 0.22±0.01 | 0.26±0.01 | 0.23±0.01 | 0.25 ±0.01 | 0.19±0.01 |
| ENT | 1.44±0.08 | 1.37±0.01 | 0.93±0.04 | 2.55± 0.10 | 1.35±0.02 | 2.23±0.08 | 3.08±0.01 | 2.97±0.04 |
| ENB | 1.65±0.06 | 1.73±0.01 | 2.02±0.04 | 1.18±0.10 | 1.71±0.03 | 1.16±0.06 | 1.08±0.01 | 1.18±0.06 |

Table 15: Foreign Exchange 2018

| Metrics | A2C | PPO | SAC | SARL | DeepTrader | EIIE | IMIT | AlphaMix+ |
|---|---|---|---|---|---|---|---|---|
| TR(%) | -5.43±2.29 | -5.25±2.45 | -5.61±3.02 | -4.52±1.91 | -4.99±2.82 | -6.15±0.28 | -5.53± 0.05 | -4.92±0.32 |
| SR | -1.01±0.45 | -0.97±0.49 | -0.91±0.43 | -0.93±0.41 | -0.91±0.55 | -1.15±0.14 | -0.97± 0.01 | -1.15±0.10 |
| CR | -0.50±0.14 | -0.48±0.16 | -0.50±0.18 | -0.49±0.19 | -0.49±0.21 | -0.59±0.04 | -0.49± 0.01 | -0.57±0.03 |
| SoR | -1.43±0.64 | -1.36±0.68 | -1.23±0.44 | -1.42±0.62 | -1.29±0.74 | -1.66±0.26 | -1.5± 0.02 | -1.77±0.16 |
| MDD(%) | 10.4±1.67 | 10.4±1.85 | 10.7±1.80 | 9.08±0.96 | 9.41±2.20 | 10.6±0.91 | 11.2± 0.05 | 8.66±0.14 |
| VOL(%) | 0.34±0.02 | 0.34±0.02 | 0.37±0.03 | 0.31±0.02 | 0.34±0.01 | 0.34±0.03 | 0.25± 0.0 | 0.27±0.01 |
| ENT | 1.38±0.03 | 1.34±0.03 | 0.94±0.03 | 2.23±0.37 | 1.37±0.01 | 1.55±0.72 | 3.08± 0.01 | 2.98±0.03 |
| ENB | 1.45±0.01 | 1.46±0.03 | 1.65±0.05 | 1.16±0.09 | 1.45±0.02 | 1.39±0.27 | 1.08±0.01 | 1.11±0.01 |

Table 16: Foreign Exchange 2017

| Metrics | A2C | PPO | SAC | SARL | DeepTrader | EIIE | IMIT | AlphaMix+ |
|---|---|---|---|---|---|---|---|---|
| TR(%) | 6.93±1.71 | 6.85±1.44 | 8.25±3.62 | 5.81±2.26 | 5.94±2.50 | 6.71±2.72 | 6.42± 0.16 | 7.16±0.36 |
| SR | 1.34±0.29 | 1.32±0.24 | 1.41±0.48 | 1.34±0.57 | 1.13±0.51 | 1.26±0.47 | 0.81± 0.02 | 1.72±0.13 |
| CR | 0.82±0.12 | 0.81±0.12 | 0.88±0.17 | 0.79±0.27 | 0.77±0.19 | 0.88±0.11 | 0.73± 0.01 | 0.93±0.02 |
| SoR | 2.15±0.45 | 2.14±0.42 | 2.17±1.01 | 2.28±0.97 | 1.74±0.89 | 1.89±0.80 | 1.21± 0.03 | 3.04±0.27 |
| MDD(%) | 8.21±1.28 | 8.20±1.12 | 9.01±2.45 | 7.12±1.27 | 7.46±1.87 | 7.33±2.08 | 8.98± 0.12 | 7.51±0.43 |
| VOL(%) | 0.32±0.01 | 0.32±0.01 | 0.35±0.03 | 0.28±0.04 | 0.33±0.02 | 0.34±0.08 | 0.36± 0.01 | 0.25±0.01 |
| ENT | 1.34±0.02 | 1.35±0.02 | 0.90±0.03 | 2.17±0.45 | 1.37±0.01 | 1.41±0.57 | 3.08±0.01 | 2.98±0.10 |
| ENB | 1.58±0.01 | 1.59±0.04 | 1.82±0.02 | 1.29±0.15 | 1.57±0.02 | 1.55±0.18 | 1.05±0.04 | 1.10±0.04 |

## C.2 Ablation Studies

| Models | Ensemble | Weight BB | Diveristy | TR(%)↑ | SR↑ | CR↑ | SoR↑ | MDD(%)↓ | VOL(%)↓ | ENT↑ | ENB↑ |
|---|---|---|---|---|---|---|---|---|---|---|---|
| SAC | | | | 7.77 | 0.37 | 0.30 | 0.50 | 42.8 | 2.45 | 1.61 | 1.10 |
| AlphaMix+ | √ | | | 10.2 | 0.46 | 0.50 | 0.55 | 31.6 | 2.15 | 3.25 | 1.01 |
| | √ | √ | | 10.1 | 0.45 | 0.48 | 0.54 | 32.3 | 2.15 | 3.28 | 1.01 |
| | √ | | √ | 10.9 | 0.47 | 0.52 | 0.57 | 32.0 | 2.19 | 3.26 | 1.02 |
| | √ | √ | √ | 12.7 | 0.52 | 0.47 | 0.62 | 38.2 | 2.16 | 3.22 | 1.00 |

Table 17: US Stock 2020 Ablation

| Models | Ensemble | Weight BB | Diveristy | TR(%)↑ | SR↑ | CR↑ | SoR↑ | MDD(%)↓ | VOL(%)↓ | ENT↑ | ENB↑ |
|---|---|---|---|---|---|---|---|---|---|---|---|
| SAC | | | | 13.4 | 0.59 | 0.41 | 0.71 | 29.9 | 0.81 | 1.10 | 1.27 |
| AlphaMix+ | √ | | | 13.6 | 0.75 | 0.77 | 0.83 | 19.0 | 0.69 | 3.15 | 1.01 |
| | √ | √ | | 11.6 | 0.66 | 0.68 | 0.73 | 18.7 | 0.68 | 3.09 | 1.02 |
| | √ | | √ | 12.0 | 0.68 | 0.70 | 0.76 | 18.7 | 0.68 | 3.20 | 1.02 |
| | √ | √ | √ | 14.8 | 0.79 | 0.53 | 0.89 | 29.1 | 0.69 | 3.12 | 1.02 |

Table 18: China Stock 2020 Ablation

| Models | Ensemble | Weight BB | Diveristy | TR(%)↑ | SR↑ | CR↑ | SoR↑ | MDD(%)↓ | VOL(%)↓ | ENT↑ | ENB↑ |
|---|---|---|---|---|---|---|---|---|---|---|---|
| SAC | | | | 16.0 | 0.51 | 0.59 | 0.59 | 55.7 | 4.46 | 0.45 | 1.07 |
| AlphaMix+ | √ | | | 24.4 | 0.73 | 0.72 | 0.76 | 44.8 | 3.26 | 2.08 | 1.00 |
| | √ | √ | | 27.7 | 0.77 | 0.78 | 0.82 | 46.5 | 3.42 | 2.14 | 1.00 |
| | √ | | √ | 32.2 | 0.83 | 0.86 | 0.89 | 48.0 | 3.62 | 2.19 | 1.00 |
| | √ | √ | √ | 33.8 | 0.86 | 0.87 | 0.94 | 46.5 | 3.46 | 2.17 | 1.00 |

Table 19: Crypto 2020 Ablation

| Models | Ensemble | Weight BB | Diveristy | TR(%)↑ | SR↑ | CR↑ | SoR↑ | MDD(%)↓ | VOL(%)↓ | ENT↑ | ENB↑ |
|---|---|---|---|---|---|---|---|---|---|---|---|
| SAC | | | | -1.02 | -0.21 | -0.02 | -0.26 | 6.94 | 0.31 | 0.93 | 2.02 |
| AlphaMix+ | √ | | | 1.06 | 0.33 | 0.26 | 0.58 | 4.24 | 0.21 | 2.98 | 1.22 |
| | √ | √ | | 1.11 | 0.34 | 0.27 | 0.60 | 4.47 | 0.22 | 2.91 | 1.19 |
| | √ | | √ | 1.04 | 0.33 | 0.25 | 0.57 | 4.34 | 0.21 | 3.01 | 1.19 |
| | √ | √ | √ | 1.22 | 0.41 | 0.36 | 0.74 | 3.77 | 0.19 | 2.97 | 1.18 |

Table 20: Foreign Exchange 2019 Ablation

## C.3  Parameter Analysis: Probing Sensitivity

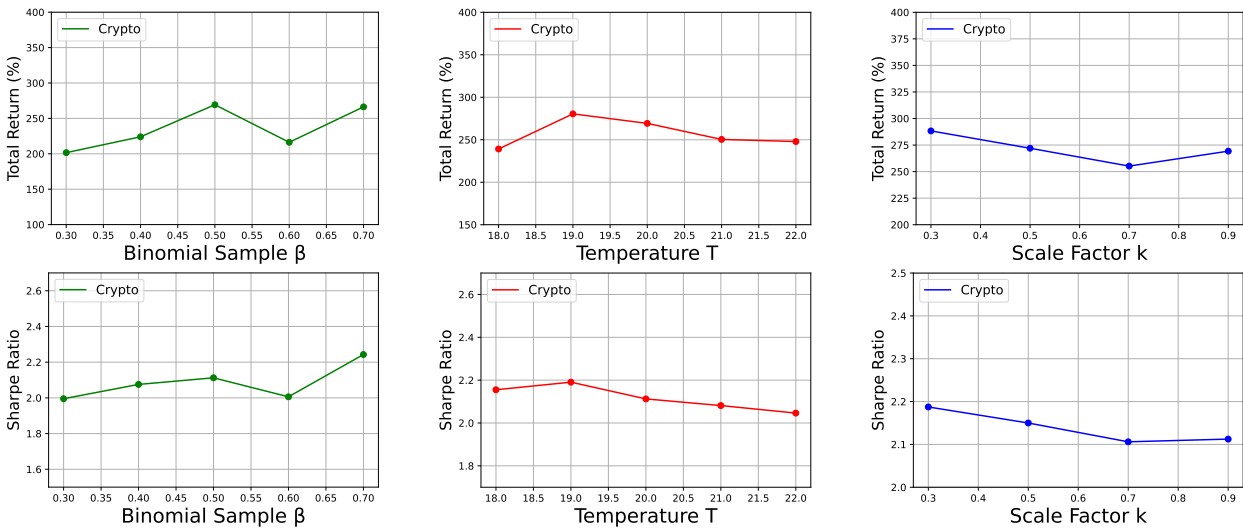

Figure 10: Hyperparameter Sensitivity Experiment Results on Crypto

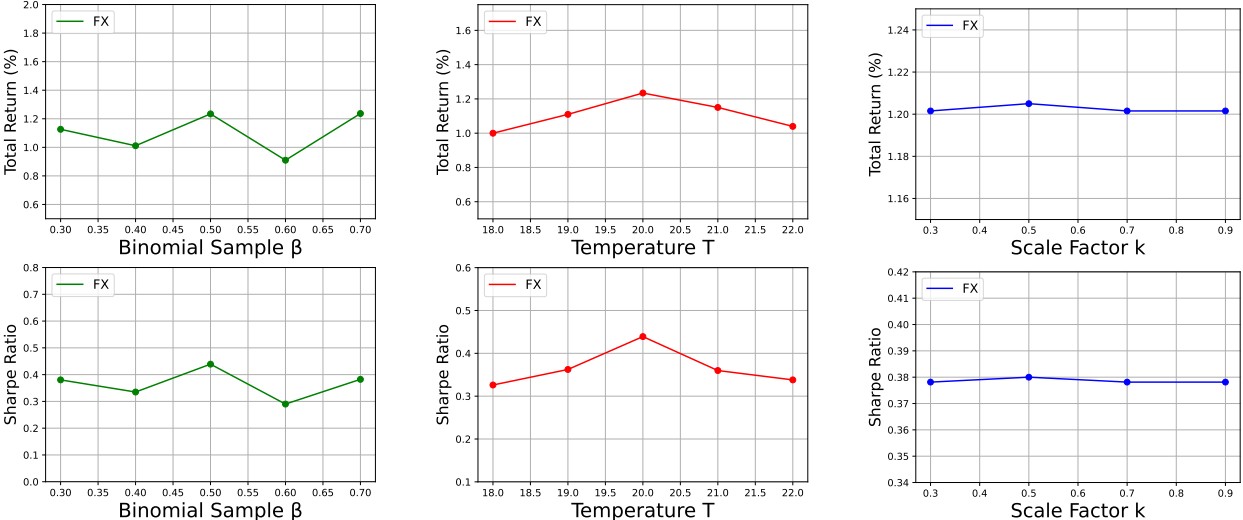

Figure 11: Hyperparameter Sensitivity Experiment Results on FX

## C.4 PRIDE-Star

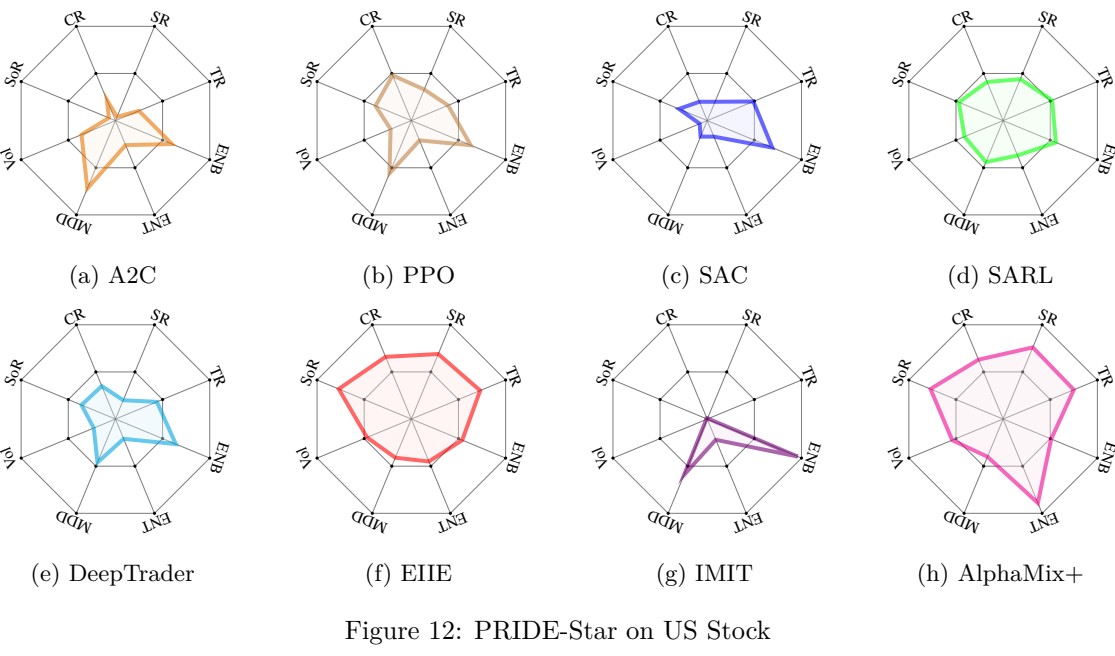

Figure 12: PRIDE-Star on US Stock

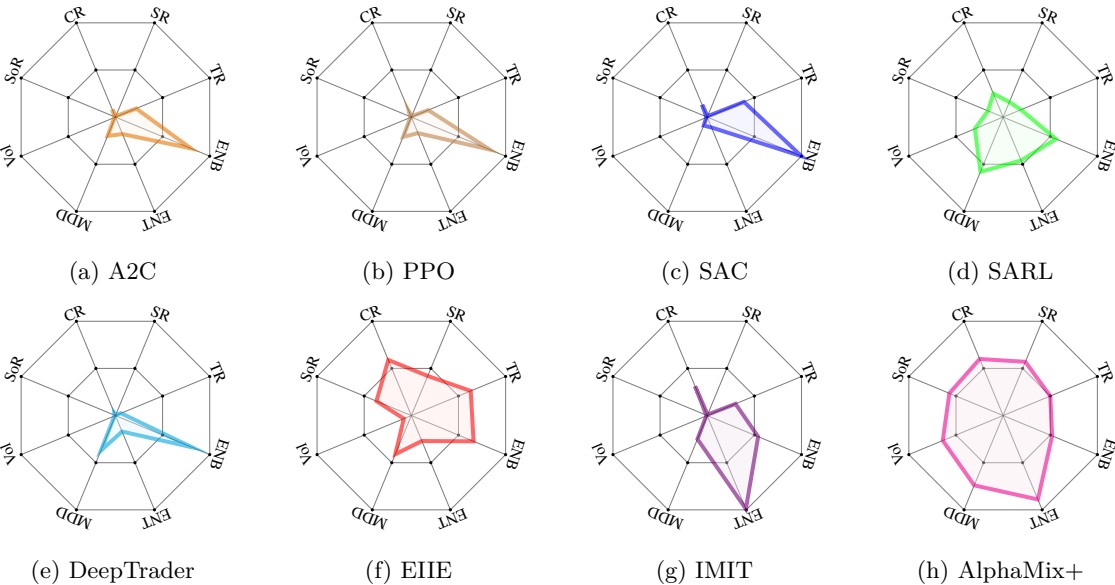

Figure 13: PRIDE-Star on FX

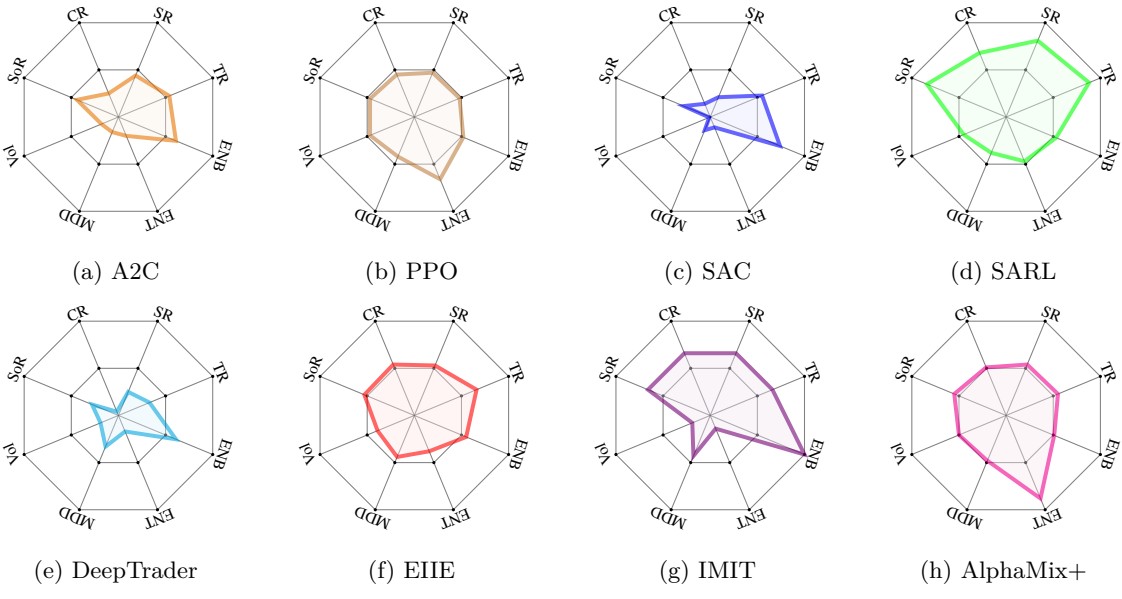

| (a) A2C | (b) PPO | (c) SAC | (d) SARL |
| --- | --- | --- | --- |
| (e) DeepTrader | (f) EIIE | (g) IMIT | (h) AlphaMix+ |

Figure 14: PRIDE-Star on China Stock

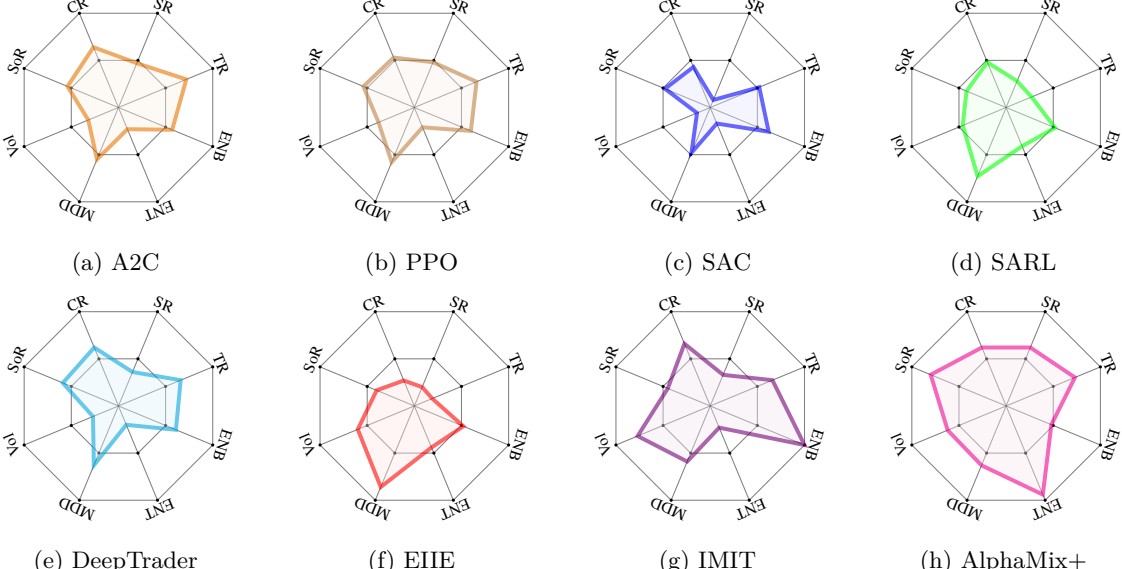

| (a) A2C | (b) PPO | (c) SAC | (d) SARL |
| --- | --- | --- | --- |
| (e) DeepTrader | (f) EIIE | (g) IMIT | (h) AlphaMix+ |

Figure 15: PRIDE-Star on Crypto

## C.5 Performance Profile

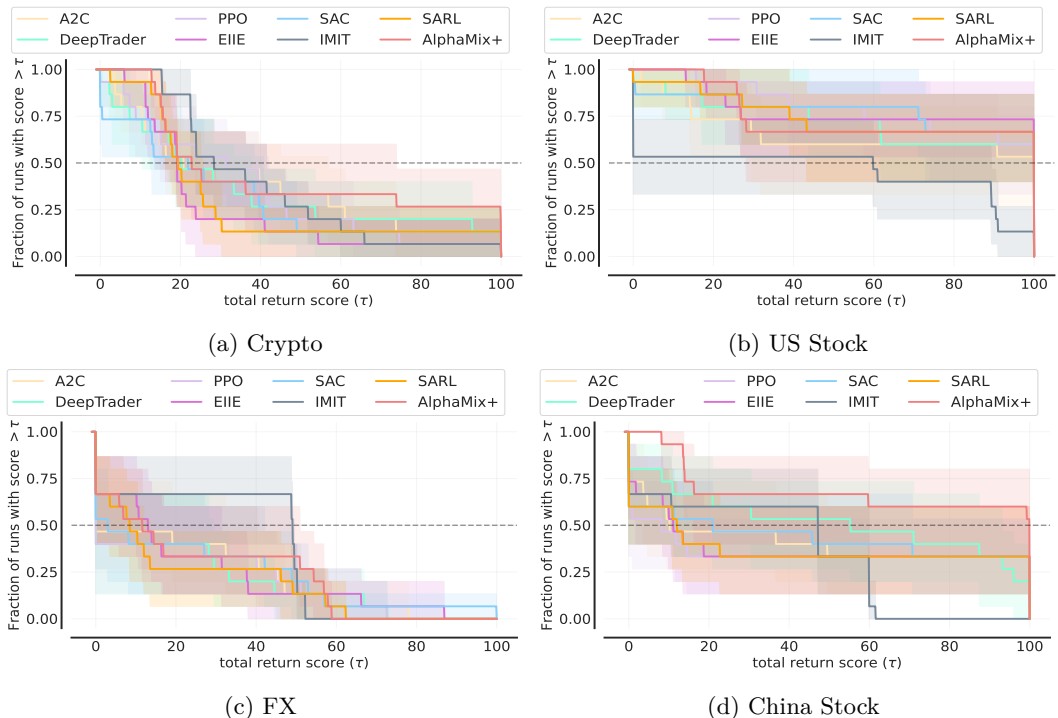

Figure 16: Performance profile on 4 influential financial markets

## C.6 Rank Distribution

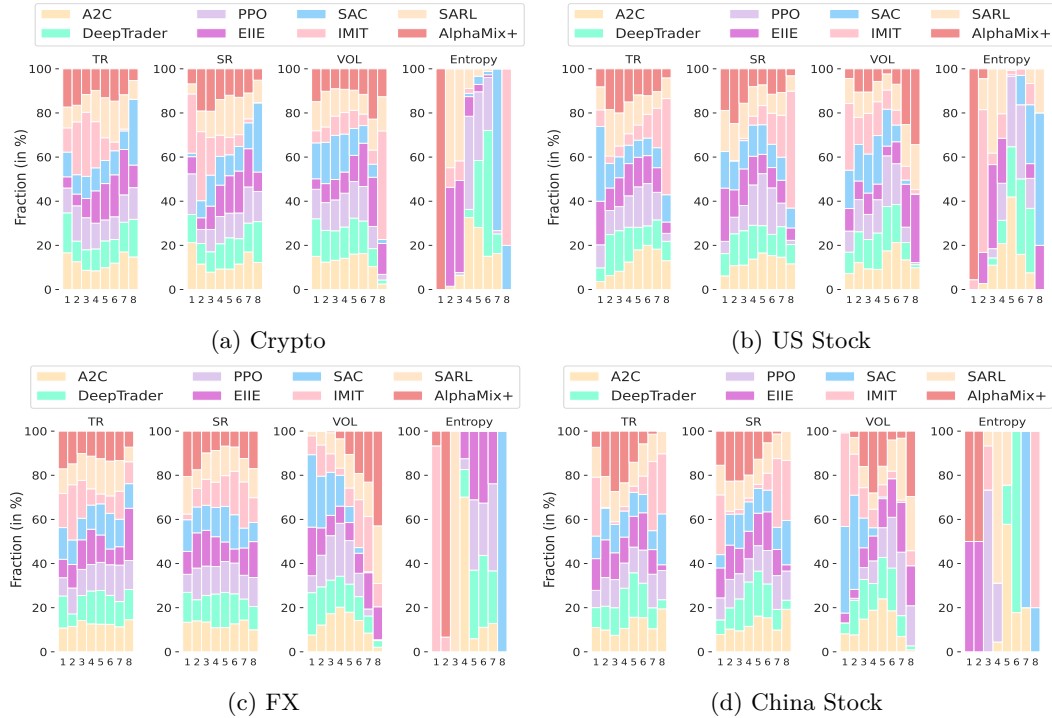

Figure 17: Rank distribution on 4 influential financial markets

