# OpenReview forum: "PRUDEX-Compass: Towards Systematic Evaluation of Reinforcement Learning in Financial Markets"
_TMLR — Accepted by TMLR_

### Review · Reviewer_kMKe · 2022-12-09

**Summary Of Contributions:**

The paper introduces an evaluation framework for reinforcement learning (RL) methods for financial markets (FinRL), which are increasingly gaining prominence in the research and applied world. The evaluation framework put forth (called “PRUDEX-Compass”) tries to measure evaluation metrics of interest beyond profitability and returns, as standard measures do. Specifically, the evaluation axes of interest in PRUDEX are Profitability, Risk-control, Universality, Diversity, rEliability, and eXplainability. Subsequently, the paper proposes another RL baseline method for FinRL called AlphaMix+ which leverages mixture-of-experts (MoE) and risk-sensitive approaches to make diversified risk-aware investment decisions. The paper evaluates AlphaMix+ and 7 other FinRL approaches on 4 real-world financial market datasets and reports the results on the PRUDEX-compass. The paper also provides a latex (tikzpictures) implementation for PRUDEX and a python package for creating the latex code from data. The paper finally discusses other, orthogonal approaches from the literature on standardizing evaluation of RL methods and the potential impacts of PRUDEX-compass.


**Audience:**

Yes

**Broader Impact Concerns:**

I think the broader impact of the paper is adequately addressed

**Claims And Evidence:**

Yes

**Requested Changes:**

1) I am not convinced that the “compass” architecture of the visualization is required. I agree that there needs to be a standardized reporting system, I’m just not convinced that the one put forth by the paper is the correct/best one.

2) Make sure the “story” is complete, based on the guidelines and questions I posed above.

3) Strengthen the formalisms of the paper.


**Strengths And Weaknesses:**

[Strengths and Weaknesses]

[+] From what I am aware, prior to this work there was no standard for evaluating RL methods for financial markets. So this paper is making a first step toward a framework for standardized reporting of metrics that matter for financial market evaluation algorithms.

[+] I can tell that the authors made significant efforts to think about measures of interest for the evaluation framework.

[+] I really appreciated the authors' efforts to make the visualization materials available through the latex code and the python package. One minor suggestion here is to have colors that are friendly for color-blind folks.


[-] The story of the paper is fundamentally about the PRUDEX-compass framework. So I don’t understand why the authors chose to include a baseline algorithm (AlphaMix+) too. It takes away from the rest of the paper’s story. Also related to the paper’s story, I didn’t understand the difference between PRUDEX-compass and PRIDE-star. It looks to me (but I may be wrong) that the PRIDE-star is part of the PRUDEX-compass but I may be wrong. It should have been made much clearer to the reader how the two are connected.

[-] I’m sorry to say this, but I don’t see why the PRUDEX-compass is the best visualization tool for the purposes that the paper describes; I find it quite confusing and unnecessarily complicated at times. Why didn’t the authors choose to report the relevant metrics (and their respective measurements) in just a google doc? Or something similar to the “Datasheets for Datasets” paper? Here is an example of something I find confusing about the current presentation: For the inner circle, why do the points on the different axes need to be connected and form a convex hull? Do these lines connecting the different points represent anything? From what I understand (although I may be wrong and please do correct me if so) the measurements on the axes are just the dots.

[-] In various parts, it is not clear what exactly is one of the paper’s contributions and what is already known in the literature. For example, for the metrics introduced in Sec 4.1 which of these have already appeared in the literature? If all of them previously have appeared then the paper’s contribution is to create an aggregate visualizer for them. How easy/hard is it to estimate each of these metrics in real-life datasets? Another example: is the t-SNE visualization something that the paper introduced or was this already known? (Tangentially related here: I didn’t understand what the point of talking about the t-SNE visualization was at this point. Were the authors trying to say that t-SNE is a data visualization but that PRUDEX-compass also includes algorithmic considerations?)

[Additional Comments]

* The formalism in the paper needs improvement at various points. For example, in Sec 2.2 T is used to denote both the time horizon and the transaction function. At equation (3), there is a different notation used (Q_\theta) as opposed to the one introduced before (Q^\pi). In the “Universality” definition (Sec. 4) how is “decent” performance defined formally?

* In Sec. 4.2 I don’t understand what the point of having the paragraph “FinRL Explainability” is since above you have talked about “explainability” again. Is there a difference between general explainability and FinRL explainability?

* In the definition of OHLCV what is function f_k?

[Typos]

* There are many acronyms that are not helpful for the reader to keep in mind. I would suggest that the authors spell out most of the words.

* Pg 2: “also provide the RL community” → “also provides the RL community”
* Pg 2: “to all QT tasks,” → “to all QT tasks.”
* Pg 3: “indicate features” → “indicates features”
* Pg 6: “Crpyto” → “Crypto”
* Pg 8: “bootrap initialization” → “bootstrap initialization”
* Pg 20: “here is the “ → this sentence abruptly ends there.

---

> ### Author Response · Authors · 2022-12-13
> **Response to Reviewer kMKe**
>
> We thank the reviewer for the valuable feedback. As the reviewer suggested, we revise the paper (orange part) and respond to questions as follows:
>
> **Q1. I don’t understand why the authors choose to include a baseline algorithm AlphaMix+. It takes away from the main story of the paper, which is systematic evaluation of RL methods in financial markets.**
>
> We observed that existing FinRL algorithms perform poorly under systematic evaluation (two examples in line 46-49) in preliminary experiments as they are designed to maximize profits only. We propose AlphaMix+ as a strong baseline with superior performance on all 6 evaluation axes. One major goal of PURDEX-Compass is to facilitate the design of novel FinRL algorithms and AlphaMix+ is in line with it. In addition, we show the usage of many evaluation toolkits by comparing AlphaMix+ with 7 existing FinRL algorithms as in Section 6.
>
> ---
> **Q2. The connection between PRUDEX-Compass and PRIDE-star is confusing. Further description is required.**
>
> In Section 5, We propose many visualization toolkits with a focus on some evaluation axes or measures in PRUDEX-Compass. PRIDE-Star is one of them involving 8 point-wise financial metrics. We include a brief description of each toolkit to clarify their usage and relationship with PRUDEX-Compass in line 330-339.
>
> ---
> **Q3. Why the PRUDEX-compass is the best visualization tool for the purposes that the paper describes? I find PURDEX-Compass unnecessarily complicated at times.**
>
> For the unnecessarily complicated concern, we clarify based on four points: i) The goal of PRUDEX-Compass is to provide a comprehensive evaluation benchmark for all kinds of investors in financial markets. To keep completeness, all axes and measures in both inner and outer layer are necessary for a comprehensive evaluation. ii) In addition to PRUDEX-Compass, we also provide many simple and easy-to-use toolkits where users can flexibly pick for the evaluation axes they care about. iii) PRUDEX-Compass provides a nice visualization tool to provide users an intuitive impression on FinRL algorithms, which is of high value for interdisciplinary research like FinRL. iv) Source code for generating PRUDEX-Compass is publicly available. Users can easily generate PRUDEX-Compass with one line of code.
>
> ---
> **Q4: Why not the authors report the relevant metrics in just a google doc? Or something like the “Datasheets for datasets” paper?**
>
> Although existing FinRL methods use tables or google docs containing numeric values of different financial metrics to show experimental results of FinRL algorithms, there are three disadvantages: i) it is hard for readers to get the rank of multiple algorithms with only pure numeric value as the value of these metrics are in different scales, ii) Investors can have quite different preferences of trading strategies. With PRUDEX-Compass, they can simply ignore the axes they do not care. For instance, aggressive traders may want to ignore risk-control and pay more attention on profitability. It is hard to achieve this with a google doc. iii) some evaluation measures such as t-SNE plot cannot be represented as a numeric value in a google doc.
>
> The objective of datasheets is to provide a standard documentation for datasets, which is not in line with the goal of this work to provide systematic evaluation. PRUDEX-Compass provides an intuitive way for investors to compare different FinRL algorithms’ performance from multiple perspectives in a systematic way. Details of contributions, design principles and advantages of PRUDEX-Compass are in line 74-84, 82-91.
>
> ---
> **Q5. For the inner circle, why do the points on different axes to be connected to form a convex hull? As I understand, the measurements on the axes are just the dots.**
>
> As claimed in Section 2.1, forming a convex hull like the inner circle can provide users an illustrative impression on different FinRL algorithms’ relative performance score of all different axes. Many impactful works (e.g., BEiT-3 [1] and iTransplant [2]) use similar plots to report experimental results from multiple perspectives.
>
> [1] Wang et al. Image as a foreign language: BEiT pretraining for all vision and vision-language tasks. Arxiv. 2022.
>
> [2] Qin et al. Closing the loop in medical decision support by understanding clinical decision-making: A case study on organ transplantation. Neural Information Processing Systems, 2021.

---

> ### Author Response · Authors · 2022-12-13
> **Response to Reviewer kMKe**
>
> **Q6. The contributions of PRUDEX-Compass are unclear. Are these measures proposed by the authors or already exist in the literature? If exists in the literature, this paper’s contribution is to create an aggregate visualizer for them. How easy/hard is it to estimate these metrics in real-life datasets?**
>
> The contributions of PURDEX-Compass are three-fold: i) carefully collecting 17 measures from the literature of multiple disciplines (e.g., finance, AI, statistics and engineering) and properly categorized them into 6 axes; ii) proposing customized version of the measures suitable for FinRL together with a set of easy-to-use visualization tools; iii) introducing PRUDEX-Compass, a unified visual interface, to show the relative performance strength of FinRL methods (inner) and their evaluation completeness (outer).
>
> We would like to emphasis the novelty of our work by a prominent example. Rliable [3], an evaluation benchmark that wins the outstanding paper award of NeurIPS 2021, also provides 5 measures (all available in existing literature) to evaluate the reliability of RL algorithms and soon become the standard protocol in RL community (135 citations). Compared to Rliable, PRUDEX-Compass provides a somehow similar but more comprehensive evaluation framework with a focus on the intersection of RL and financial markets.
>
> [3] Agarwal et al. Deep Reinforcement Learning at the Edge of the Statistical Precipice. Neural Information Processing Systems. 2021.
>
> ---
> **Q7. I didn’t understand the point of talking about t-SNE visualization at Section 5.1. Were the authors trying to say that t-SNE is a data visualization tool but PRUDEX-Compass also includes algorithmic consideration?**
>
> t-SNE [4] is a good visualization tool to map datapoints in high dimensional space to a 2-D dimension, which reveals the similarities of datapoints. In this paper, we apply t-SNE to visualize data-level diversity on temporal financial time series data as one toolkit proposed together with PRUDEX-Compass. We include descriptions of the visualization toolkits and map them with related evaluation measures of PRUDEX-Compass in line 330-339.
>
> [4] Vander et al. Visualizing data using t-SNE. Journal of Machine Learning Research. 2008.
>
> ---
> **Q8. Why the authors talk about explainability in both Section 2.1 and 2.2. What are the differences?**
>
> In Section 2.1, we highlight the importance of evaluating explainability in FinRL from a qualitative perspective. In Section 2.2, we conduct literature review of current RL explainability research and point out their potential usage as concrete measures for FinRL.
>
> ---
>
> **Q9. It is better to have colors that are friendly for color-blind folks.**
>
> We include extra color options in the Github repo for color-blind folks available at [here](https://anonymous.4open.science/r/PRUDEX-Compass-948C/Compass/generate/compass/README.md)

---

> ### Comment · Reviewer_kMKe · 2023-01-19
> **Response to rebuttal**
>
> Thank you for responding to my comments! In my opinion, the revised version of the paper is significantly improved.
>
> I want to follow up on some of the points the authors brought up in their response.
>
> **Q1**: I understand better now why you proposed this new algorithm. But in my opinion this result reads as "detached" from the rest of the paper. The story of the paper is around PRUDEX as a standardization framework for reporting. So I maintain my view that the new algorithm doesn't feel connected to the rest of the paper. It feels more like 2 results combined in 1 paper.
>
> **Q2**: So if I understand correctly, PRIDE-star is another, more specialized visualization toolkit. Is that so? There is also a typo in line 331: "genral" --> "general"
>
> It is still my personal opinion that PRUDEX-Compass is rather complicated, but I do see the authors' points that there are a lot of measures of interest for FinRL.

---

> > ### Author Response · Authors · 2023-01-20
> > **Response to Reviewer kMKe**
> >
> > Thank you for the positive attitude on the revised version and timely feedback. Our responses are as follows:
> >
> > **Q1. I understand better now why you proposed this new algorithm. But in my opinion this result reads as "detached" from the rest of the paper. So I maintain my view that the new algorithm doesn't feel connected to the rest of the paper.**
> >
> > Besides the points mentioned above, we would like to further clarify with the following three points:
> >
> > * We emphasis the advantages of introducing both new algorithms and evaluation metrics in one work by an influential example [1] (6432 citations). This work proposes both a new update rule for GANs training and introduce FID as a new evaluation metric. FID helps to show the advantages of the proposed update rule and the authors leverage the proposed update rule to demonstrate the superiority of  FID over previous evaluation metrics. In our case, PRUDEX-Compass and AlphaMix+ share similar situations as [1].
> > * In the revised version, we bring Section 2 (e.g., contents of PURDEX-Compass) forward to help readers focus on the main story of this work. We also  include a footnote at the start of Section 4 (e.g., contents of AlphaMix+) to remind readers whose main interests lie in the evaluation benchmark that they can skip that section and take AlphaMix+ as a strong FinRL algorithm.
> > * AlphaMix+ can be considered as a side contribution, which makes this paper stronger with contributions from both evaluation and algorithm perspectives.
> >
> > [1] Heusel et al. GANs trained by a two time-scale update rule converge to a local Nash equilibrium. Neural Information Processing Systems. 2017.
> >
> > ---
> > **Q2. if I understand correctly, PRIDE-star is another, more specialized visualization toolkit. Is that so? There is also a typo in line 331.**
> >
> > Yes, your understanding is correct.  PRIDEX-star is a more specialized visualization toolkit, which can be used independently to evaluate measures belonging to the profitability, risk and diversity axes in PRUDEX-Compass.
> >
> > Thanks for the notice. We have fixed the typo in the revised version.
> >
> > ---
> > **Q3. It is still my personal opinion that PRUDEX-Compass is rather complicated, but I do see the authors' points that there are a lot of interest for FinRL.**
> >
> > Besides the above 4 points mentioned in our responses to Q3, we could like to share some comments and feedback of PRUDEX-Compass from our industry collaborators. We hope their positive attitude after using PRUDEX-Compass can address the "too complicated" concern.
> >
> > The first feedback is a junior quantitative researcher in a brokerage firm. He told us that his daily work involves writing reports to comprehensively analyse the performance of different FinRL algorithms. He tried to generate the PRUDEX-Compass and other plots with our scripts and visualization tools, and find them very helpful.
> >
> > The second feedback is from a portfolio manager, who has  over 8 years quantitative trading experience and currently work in a medium size hedge fund. He said that systematic evaluation is obvious of vital importance for quantitative trading methods and most evaluation measures in PRUDEX-Compass is in line with their taste on algorithm evaluation. They will try to evaluate their algorithms with PRUDEX-Compass in the future as it is comprehensive with user-friendly interface and source code.

---

### Review · Reviewer_NTff · 2022-12-15

**Summary Of Contributions:**

The paper looks at providing a holistic evaluation for RL methods on financial markets. Several RL baselines are considered and their performance along different aspects (reliability, diversity,  risk control, etc, ) are provided. The core contribution is identification of relevant metrics and providing a variant of the star plot to visualise these results. EMpirical results are provided on data from 4 financial markets.

Overall, there are three key aspects in the paper, (a) what are the metrics to be considered, (b) what are the performances of the baselines on those metrics, and (c) how to report them in a visually appealing manner. The paper tries to tackle everything, but unfortunately, this makes the paper unclear/incomplete on all of those aspects. Isee Lines, It is not clear what the


**Audience:**

Yes

**Broader Impact Concerns:**

-

**Claims And Evidence:**

No

**Requested Changes:**

1. Since this is a domain-specific paper, can the instantiation of the RL setup be better discussed? For instance, what are the actions here? What is the time horizon? What corresponds to an episode? Is there a discounting factor? What is the reward function (even after I finished reading the paper it was not clear to me what is the task for the RL agent here)? Is it purely offline RL or there was a model built using the past data on which online RL was done? What part of the data was used for training and on what part was the performance evaluated?


2. Explain the weighting in Eq 7. My understanding is that the weight should be smaller when the Q functions disagree, i.e., the ensemble has a higher standard deviation, and because of this Eqn 7 uses negative $\bar Q_{std}$. However, I am not sure about Line 145. Since $\bar Q_{std}$ is always positive, wouldn’t $\sigma( - \bar Q_{std})$ be between [0, 0.5], and therefore the range would be [k, 0.5 +k], instead of [0.5, 0.5+k]? Maybe I am misunderstanding something?


3. How to select $T$ and $k$ in Eqn 7 during experiments? How to choose $\beta$?


4. Line 156, $L_\pi$ is not defined. I am assuming it is similar to $L_{actor}$, but I am not sure which Q value is being used here from the ensemble. Could the authors clarify this?


5, Can authors convert Lines 165 to 169 into equations for better clarity about what is the exact procedure? I think I get what the procedure is trying to do, but again many words have been used very loosely. For instance, Line 167 says that the UCB is being optimized, but my guess is that only a _heuristic_ for the UCB is being optimized, and not necessarily the actual UCB.



PRUDEX compass

6. Figure 1 suggests that AlphaMix+ gets the best performance because of the *largest area in the inner level*. My understanding is that the area of the inner level is *not invariant to the ordering of the axes,* i.e., if one plots the axes in a different order, one might get a different area enclosed. If this is true, then authors might need to really rethink such claims.


7. Normalization discussed in lines 191-192 needs a lot more justification. Appendix A.2 is largely focused on the procedure done to obtain the plot, but not much is discussed why that procedure is meaningful. Given that this is one of the core contributions of the paper, and that different normalization schemes can potentially result in vastly different plots, the paper would benefit a lot by having a thorough discussion on this. Also, it would be helpful if the authors could elaborate on what is meant by ‘market average’ here.


8. I do not have a finance background, so I am not sure how are the values for the axes being computed. Can authors provide or point me to the exact equations for computing these? Further, can authors also comment on other axes that might be of interest to be added in the future? My first thought was that there would be something about assumptions as well, for instance, the RL setup considered in the work has a very strong assumption of stationarity, which is likely violated in financial applications. One could argue that ‘reliability’ could capture that but it looks like right now it is simply aimed at capturing sensitivity to seed.

9. Line 273: I do not think that there is a compelling reason to not compare against non-RL method. Saying that RL methods outperform non-RL based methods, and citing past works, is not sufficient. The good part about the proposed work is the holistic evaluation, which was not done by prior works. And I can imagine all of the proposed axes equally useful for non-RL methods. I understand that incorporating these non-RL methods might not be easy, but given that the core contribution of the paper is evaluating different algorithms and presenting the results, I think the paper would be much stronger with non-RL baselines.

Other plots:
10. I am not sure if I understood what is the t-sne plot showing. Does each point on the t-sne plot correspond to a single time-step feature? If yes, how is the time-series aspect captured and how should a reader infer that from these plots?

11. Line 340, how is the ‘rank probability’ computed?

12. For the strategy diversity heatmap, the portfolio is for which time? I am guessing this plot should keep changing over time?

13. Figure 8, I do not understand these plots. What is ‘score’? How is it computed? Is a larger score better or lower?


**Strengths And Weaknesses:**

Strengths:
1. Holistic evaluation is critical for high-stake settings, and this paper takes a step toward that for the financial market.

2. Extensive open-source, practical, library for evaluating and visualizing the performance of different methods.

Weakness:
1. The problem setup is unclear (see below)

2. The evaluation metric is unclear and needs more discussion (but this might be because I am not familiar with the finance aspects). It is not clear what is the actual contribution here? Are these metrics not known in the literature? Are there any understudied nuances about the metrics that people need to be careful about?

3. Baselines are not thoroughly investigated. How is sensitivity analysis done for the parameters? How were the hyper-parameters for all the baselines chosen? How robust are the metrics to the chosen constants in the metric formulation (for example the choice of 20% in Line 712)? The proposed work would be much stronger if these were discussed in detail.

4. Visualization is ok, but I am not sure about the audience for that at TMLR. Perhaps that might be more useful at data visualization conferences? (I would defer to the AE for evaluation of interest at TMLR for this aspect of the paper)

---

> ### Author Response · Authors · 2023-01-19
> **Response to Reviewer NTff**
>
> We thank the reviewer for the comprehensive and valuable feedback. As the reviewer suggested, we revise the paper (orange part) and respond to questions as follows:
>
> **Q1. The actual contribution of PRUDEX-Compass is not clear. Are these metrics existed in the literature or proposed by the authors?**
>
> The contributions of PRUDEX-Compass are three-fold: i) carefully collecting 17 measures from the literature of multiple disciplines (e.g., finance, AI, statistics and engineering) and properly categorized them into 6 axes; ii) proposing customized version of the measures suitable for FinRL together with a set easy-to-use visualization tools (Section 6.1 to 6.7); iii) introducing PRUDEX-Compass, a unified visual interface, to show the relative performance strength of FinRL methods (inner) and their evaluation completeness (outer). We include these contents in line 75-80.
>
> ---
> **Q2. Visualization is ok, but I am not sure if this paper is of high interest for TMLR audiences.**
>
> Besides visualization, we also propose a strong FinRL algorithm, collect and customize 17 evaluation measures and open source a library containing the whole pipeline of FinRL algorithms training and evaluation. This work should be of high interest of the following TMLR audiences:
>
> * For FinRL researchers, this work provides infrastructures to build and evaluate FinRL methods in a systematic way and move one step towards real-world industry deployment.
> * For RL researchers, we introduce a new challenging scenario to evaluate novel RL algorithms in the ever-changing financial markets.
> * For Fintech researchers, this work enables exploring the potential of RL in combination with finance knowledge.
> ---
> **Q3. How were hyperparameters chosen for all baselines?**
>
> For other FinRL methods, there are two conditions: i) if there are authors' official or open-source FinRL library implementations, we apply the same hyperparameters for a fair comparison. This condition applies for A2C, PPO, SAC, SARL and DeepTrader. ii) if there are no publicly available implementations, we reimplement the algorithms and try our best to maintain consistency based on the original papers. This applies for EIIE and IMIT.
>
> ---
> **Q4. Explain the weighting range in Eqn 9. There are confusions there.**
>
> Sorry for the confusion. Your understanding is correct, and we have fixed this typo in the revised version.
>
> ---
> **Q5. How robust are the metrics to the chosen constants in the metrics formulation (e.g., 20% in line 810)?**
>
> The purpose of chosen constants is to adjust the scale of normalized score of different FinRL on each axis to make the inner-level plot more distinguishable and easier to follow. The value of constants does not influence relative performance order of different FinRL methods and has not negative impact on the robustness of PRUDEX-Compass measures.
>
> ---
> **Q6. For future directions, there might be something about stationary assumptions as financial markets are violated. One could argue that reliability could capture that, but it is simply aimed at seed sensitivity at present.**
>
> Stationary assumptions are of vital importance for violated financial markets,, we plan to add one toolkit to automatically categorise market into different styles (e.g., bull/bear) and evaluate the performance of FinRL methods under different styles. Furthermore, there can be a data-driven simulator to generate unseen stylized data for further evaluation. We include these contents in line 463-466.
>
> Current reliability axe is not only aimed at seed sensitivity but also evaluating market distribution shift across time with dataset splits under rolling time window.
>
> ---

---

> ### Author Response · Authors · 2023-01-20
> **Responses to Reviewer NTff**
>
> **Q17. The instantiation of the RL setup needs to be better discussed. What are actions? What is the time horizon? What corresponds to an episode? Is there a discounting factor? What is the reward function?**
>
> We include more details in Section 3.2 to clarify the instantiation of RL setup. For actions at time $t$, it is an $M+1$ dimension vector to represent a portfolio, which indicates the proportion of capitals invested into each financial asset at the current time step. The time horizon indicates the length of the trading period. During training, one episode corresponds to adjusting the portfolio at each time step through the whole trading periods and interacting with environment to get investment results . There is a discount factor and we set it as 0.99 in practice. The reward function is the change of portfolio value  at time $t$, where positive/negative values indicate earning/losing money, respectively.
>
> ---
>
> **Q18. For the RL environments, is it purely offline RL or there was a model built using the past data on which online RL was done?**
>
> In this work, we apply the popular portfolio management environment [1] implemented based on OpenAI Gym, which simulates online live financial markets with realistic historical offline market data according to the principle of time-driven simulations. During training, we feed observations of technical indicators as input of RL agents. RL agents generate a portfolio (action) and the environment returns the portfolio value change at each time step as reward. By interacting with the environment, the trading agents will try to derive a trading strategy with high profits. Specifically, the environment assumes the trading volume of agents is not very large and has little impact on the market. Then, it is reasonable to use offline historical financial data to build a model for reward calculation during online simulation. We include these contents in line 320-327.
>
> [1] Liu et al. FinRL: A Deep Reinforcement Learning Library for Automated Stock Trading in Quantitative Finance. International Conference on AI in Finance. 2021.
>
> ---
>
> **Q19. What are the equations for computing the values of axes in PRUDEX-Compass? A thorough discussion on how and why the calculation and normalization is done in that way is necessary?**
>
> We normalize the performance of different measures on the 6 axes to a score from 0 to 100. The basic design principle is to propose a distinguishable and robust way to show the performance difference of FinRL methods across multiple evaluation measures in terms of each axe. We include related computing equations and discussion in Appendix A.3.

---

> ### Comment · Reviewer_NTff · 2023-02-03
> **Response to rebuttal**
>
> Thanks for the updates, it has helped clarify most of my concerns.
>
> 1. Metrics: As the authors mentioned, normalization does not change the relative order between the methods. However, it does change the scale at which  the performances are being visualized and the area that each of the methods cover in the preudex compass. Different choices of normalization factors can make two algorithms look visually as if they are performing very similarly or it can enable `zooming in' to better differentiate performance of different methods. Elaborating on this point is important because one big aim of the paper is to have the prudex-compass tool that can enable quickly visualizing the pros and cons of different methods.
>
> See Section 4.1 in the work by Jordan et al. (2020) for a more in-depth discussion on the considerations regarding normalization.
>
> Jordan, Scott, et al. "Evaluating the performance of reinforcement learning algorithms." International Conference on Machine Learning. PMLR, 2020.
>
> 2. Hyper-parameters: As authors mentioned, default hyper-parameters were chosen for the baseline methods. Note that this can be significantly disadvantageous for baseline methods that are tuned mostly on MujoCo type of domains and not FinRL domains. In comparison, the results for the proposed method is based on hyper-parameter tuning on the considered FinRL domains. Making this clear in the paper would be helpful.
>
> 3. Experiments: Despite the clarification, I am still confused about how the 4 historical datasets were used vs how was Liu et al.'s portfolio management online simulator used? I suspect the answer is in this sentence: "Then, it is reasonable to use offline historical financial data to build a model for reward calculation during online simulation. " (Line 320-327) Unfortunately, I do not understand this. Consider elaborating this in the draft.
>
>
> Minor: For future comments, it would be helpful if the reviewer's comments are not paraphrased and the order numbering of the questions are not shuffled. Paraphrasing/re-ordering requires more effort to pinpoint which points were and were not addressed.

---

> > ### Author Response · Authors · 2023-02-06
> > **Response to Reviewer NTff**
> >
> > Thanks for your positive attitude on the revised draft and valuable feedback. We are glad that our updates addressed your concerns. Our responses are as follows:
> >
> > **Q1. Metrics: Elaborating on choices of normalization factors is important because one big aim of the paper is to have the PRUDEX-Compass tool that can enable quickly visualizing the pros and cons of different methods.**
> >
> > We elaborate on our choices of normalization factors in line 89-99: Inspired by the widely used performance ratio in Arcade Learning Environment [1, 2], we choose to normalize FinRL methods' performance based on their relative performance to market average strategy, which is considered as the golden standard for many financial practitioners. Specifically, we assign score $t=50$ to market average and introduce a parameter $k$%, where $k$% better or worse relative to the market average is assigned 100 and 0, respectively. We further clip score values, which are larger than 100 to 100 and lower than 0 to 0, to alleviate the impact of extreme values (e.g., extremely high return rate under markets with sudden increase of some Crypto assets). For the value choice of $k$, we find it is an empirically robust option to set $k=20$ for our experiments on 8 FinRL methods across 4 financial markets. Moreover, we consult with our industry collaborators and they agree that it is reasonable to consider 20\% better than the market average as a very successful trading strategy (e.g., $t=100$).
> >
> > In addition, we find the discussions in [3] very insightful. To further improve the accuracy and reliability of PRUDEX-Compass, we plan to customize the scientific and non-exploitative RL evaluation procedures in [3] into FinRL domains for the normalization and summarization of experimental results in the future. Related contents are included in line 484-486.
> >
> > [1] Mnih et al. Human-level control through deep reinforcement learning. Nature. 2015.
> >
> > [2] Machado et al. Revisiting the arcade learning environment: Evaluation protocols and open problems for general agents. Journal of Artificial Intelligence Research. 2018.
> >
> > [3] Jordan et al. Evaluating the performance of reinforcement learning algorithms. International Conference on Machine Learning. 2020.
> >
> > ---
> > **Q2. Hyper-parameters: As authors mentioned, default hyper-parameters were chosen for the baseline methods. Note that this can be significantly disadvantages for baseline methods that are tuned mostly on Mujoco types of domains and not FinRL domains. In comparison, the results for the proposed method is based on hyper-parameter tuning on the considered FinRL domains.**
> >
> > The default hyper-parameters for baselines we followed are from implementations in a widely used FinRL platform [4] or authors' official implementations of FinRL methods, **which are not tuned in MujoCo type of domains but FinRL domains**. We clarify this with footnote 9.
> >
> > [4] Lin et al. FinRL: A deep reinforcement learning library for automated stock trading in quantitative finance. International Conference on AI in Finance. 2021.
> >
> > ---
> > **Q3. Experiments: Despite the clarification, I am still confused about how the 4 historical datasets were used vs how was Liu et al‘ portfolio management online simulator used? I suspect the answer is in this sentence: "Then, it is reasonable to use offline historical financial data to build a model for reward calculation during online simulation. " (Line 320-327) Unfortunately, I do not understand this. Consider elaborating this in the draft.**
> >
> > We provide a concrete example here to further clarify how FinRL environment is built. Considering a simple trading scenario with only one stock, we obtain $p_t^c$ and $p_{t+1}^c$, which denote the close price of the stock at time $t$ and $t+1$, from historical data. The action at time $t$ is to buy $k$ shares of the stock. Then, the reward $r_t$ at time $t$ is the account profit defined as $k * (p_{t+1}^c-p_{t}^c)$. For state, we use historical market data to calculate technical indicators in Table 3 as external state and investors' private information such as remaining cash and current position is applied as internal state. Similar procedures to build RL environment with historical market data have been applied in many FinRL works [4, 5, 6]. Readers may check our open source code for more details of the RL environment. Related contents are included in line 339-346.
> >
> > [5] Wang et al. Deeptrader: A deep reinforcement learning approach to risk-return balanced portfolio management with market conditions embedding. AAAI Conference on Artificial Intelligence. 2021.
> >
> > [6] Ye et al.Reinforcement-learning based portfolio management with augmented asset movement prediction states. AAAI Conference on Artificial Intelligence. 2020.

---

### Review · Reviewer_95pS · 2023-01-05

**Summary Of Contributions:**

This paper aims to provide a systematic evaluation of FinRL algorithms in Financial Markets. The paper propose PRUDEX-Compass, which includes 6 dimensional metrics,  Profitability, Risk-control, Universality, Diversity, rEliability, and eXplainability. AlphaMix+ is proposed as a strong baseline, which includes Risk-aware Bellman Backup and diversified experts. 8 FinRL methods are evaluated on 4 real-world financial datasets.

**Audience:**

Yes

**Broader Impact Concerns:**

None.

**Claims And Evidence:**

Yes

**Requested Changes:**

1.Provides detailed ablation study of AlphaMix+.
2.Specify more details on the RL environments in the experimental parts.

**Strengths And Weaknesses:**

Strengths:
1.The paper is clearly written.
2.The paper propose an evaluation framework for FinRL algorithms, which has great value for RL and FinRL community.
3.The visualization of algorithms is good.

Weaknesses:
1.The proposed AlphaMix+ algorithm lacks the ablation study, which part is the most critical for the performance?
2. The variance of AlphaMix+ is larger than baselines.
3. There seems to be 2 parameters to be tuned, which may cost much computation costs.

---

> ### Author Response · Authors · 2023-01-19
> **Response to Reviewer 95pS**
>
> We thank the reviewer for the valuable feedback. As the reviewer suggested, we revise the paper (orange part) and respond to questions as follows:
>
> **Q1. The proposed AlphaMix+ algorithm lacks the ablation study, which part is the most critical for the performance?**
>
> We conduct ablation studies on the effectiveness of each component in AlphaMix+ on all 4 datasets. As shown in Table 17-20, all three components (mixture-of-experts, risk-aware Bellman backup and tricks to encourage diversity) in AlphaMix+ have positive impact and can be fruitfully integrated together to further improve the performance.
>
> ---
> **Q2. There seems to be 2 parameters to be tuned, which may have high computation costs.**
>
> In our experiments, we tune three hyperparameters: i) temperature $T$; ii) confidence range $k$; iii) Bernoulli distribution $\beta$. The computation cost is at a reasonable level for two reasons: i) it only takes about 1-1.5 hours to fully train an RL agent on one dataset using one RTX 3090 GPU; ii) we search for proper hyperparameter values on Crypto and FX datasets and apply the same hyperparamers on China and US stock datasets to avoid over-tuning.
>
> ---
> **Q3. Specify more details on the RL environments in the experimental parts.**
>
> We include contents of RL environment assumptions, interaction loop and implementation details (line 331-346). Related informations such as MDP formulation (line 173-190), datasets (line 263-279), features (line 281-290) and training setup (line 318-329) are also included in the revised version to help readers understand the whole RL training pipeline.
>
> ---
> **Q4. The variance of AlphaMix+ is larger than baselines.**
>
> As shown in Table 5-16, the performance of AlphaMix+ is stable among different random seeds. It achieves the lowest performance variance in US stock, China stock and Crypto datasets and 3rd place in the FX dataset.

---

### Author Response · Authors · 2023-01-23
**Paper Revision Summary**

Dear three reviewers and AE,

Thanks for your insightful reviews and suggestions. We have revised the paper and look forward to your feedback. A summary of major changes is as follows:

* We restructure the paper to highlight "main story" and summarize three core contributions of PRUDEX-Compass (line 74-80).
* We include motivations of the AlphaMix+ algorithm and its relationship with the evaluation benchmark (line 9-13, 46-49, 189-193).
* Inference details of AlphaMix+ with related equations are added (line 239-248).
* We conduct experiments of AlphaMix+ for ablation studies (Table 17-20) and hyperparameter sensitivity analysis (Appendix D.3), respectively.
* We include more details on the design of PRUDEX-Compass axes (line 82-91) and measures (line 128-130, 134-137).
* Concrete instantiations of FinRL tasks and MDP formulations are available (line 139-143, 157-160, 171-178).
* We add details on data split (line 263-267), baseline descriptions (line 284-300), hyperparameter selection (line 307-317) and RL environment implementation (line 319-328).
* We describe the design details and usage of proposed visualization tools, i.e., t-SNE (line 367-369), rank distribution (line 400-406), diversity heatmap (line 413-416) and extreme market (line 434-437), and their connections with measures in PRUDEX-Compass (line 331-340).
* We discuss our future plans to improve this work in line 463-473.

Thanks again for your reviews and suggestions. We’d be happy to engage in further discussions and continue to improve this paper.

Best, Authors of Paper 598

---

### Decision · Action_Editors · 2023-02-17

**Recommendation:** Accept with minor revision

**Comment:**

This manuscript introduces a tool, dubbed PRUDEX-Compass, for evaluating reinforcement learning in financial markets (FinRL), which is most systematic and comprehensive for FinRL evaluation to the best of AE’s knowledge. The authors also proposed a strong FinRL baseline named AlphaMix+. Per the insightful reviews, the authors did a good rebuttal such that most of the problems and concerns have been well addressed. There remain several minor issues pointed out by the reviewers, for which the AE recommends Minor Revision. The authors are suggested to fix all the issues and emphasize the impact of this research on the FinTech community in the upcoming revision.

**Audience:**

Yes.

**Claims And Evidence:**

Yes.

---

> ### Author Response · Authors · 2023-02-21
> **Cemera-ready Version Updated**
>
> Thanks for your positive appreciation of the paper. We would like to thank AE and three reviewers again for their helpful comments and suggestions that have strongly improved the paper during the reviewing process.
>
> We have updated the camera-ready version with contents of impact on the FinTech community. In addition, we polish the manuscript, fix typos, reorganize Appendix and prepare open-source code to further improve the quality of the work.

---

> ### Public Comment · ~Martin_Mundt1 · 2023-03-02
> **Request for revision to give credit to the CLEVA-Compass**
>
> Dear authors, dear reviewers and editor,
>
> this paper has come to our attention recently and we would like to request a small revision to **assign appropriate credit to our earlier CLEVA-Compass** work. To be more specific, our published ICLR 2022 work "CLEVA-Compass: A Continual Learning Evaluation Assessment Compass to Promote Research Transparency and Comparability" (https://openreview.net/forum?id=rHMaBYbkkRJ) seems to be the basis for the PRUDEX-Compass. We believe this is great and are happy to see its further adoption.
>
> However, the compass itself is a direct adaptation of our work, descriptions are clearly inspired by it, and the distributed GitHub package with LaTeX + tikz support is a modified version of our open-source source. We had naturally distributed the code so others can adapt it, but with credit being given properly (https://github.com/ml-research/CLEVA-Compass). We think this should be pointed out to avoid misleading the reader and give credit where it is due, rather than having a sentence in passing in section 2.1 "We choose the design of the axis-level compass as a star diagram following the idea of (Wang et al., 2022; Mundt et al., 2021)." In addition, the arXiv version of the PRUDEX-Compass should be updated, which does not seem to cite us at all: https://arxiv.org/pdf/2302.00586v1.pdf
>
> We would like to **emphasize that we do in fact like the paper and encourage the adaptation of the CLEVA-Compass towards the PRUDEX-Compass. It's great to see versions spread from the original** continual machine learning proposal to the now financial domain. The paper has a lot to offer and has many differences to our original in the domain specific parts, so a clarification wrt inspiration and source material is sufficient. It will hopefully be an easy fix. We enjoy the remaining assessment on FinRL. **Please simply include a statement that you are inspired by and are building on top of our work prominently at the start, as well as include a statement that your disseminated code is forked directly from our template.**
>
> We are looking forward to the increased adoption of both the CLEVA and PRUDEX compass.
>
> Best,
>
> the first author of the CLEVA-Compass

---

> > ### Author Response · Authors · 2023-03-02
> > **Response to CLEVA-Compass Author**
> >
> > Dear authors of CLEVA-Compass,
> >
> > Thanks for your kindly suggestions. We fully appreciate the great work of CLEVA-Compass and make the following revisions as you mentioned:
> >
> > * At the end of Section 2's first paragraph, we remark that the visualization design of PRUDEX-Compass is inspired and built on top of the template code provided in CLEVA-Compass and list the corresponding Github repository link at the footnote.
> > * We have modified the acknowledgement part of PRUDEX-Compass Github page (https://github.com/TradeMaster-NTU/PRUDEX-Compass/) to clearly give credit of CLEVA-Compass.
> > * We will update the arxiv version soon and it may take some time to be processed by arxiv.
> >
> > In addition, we would like to highlight the __differences__ between PRUDEX-Compass and CLEVA-Compass to avoid possible confusions of PRUDEX-Compass's contributions : i) From the domain perspective, PRUDEX-Compass is designed for RL in financial markets and CLEVA-Compass is designed for continual learning; ii) From the objective perspective, PRUDEX-Compass serves as a systematic evaluation benchmark while CLEVA-Compass focuses on demonstrating the relationship of existing literatures in continual learning; iii) Since the two works are designed for different domains and objectives, the contents of both inner and outer level are fully different with __no overlap__. iv) The inner-level of PRUDEX-Compass is designed to show the relative strength of different FinRL algorithms with normalised score range in [0, 100] while the inner-level of CLEVA-Compass is designed with binary value to indicate supervised/unsupervised settings; v) Besides the compass plot, PRUDEX-Compass also include related datasets, RL environment, baseline implementations and 6 other evaluation tools (e.g., PRIDE-Star), where CLEVA-Compass does not include.
> >
> >
> > Best regards,
> >
> > Authors of paper 598